

# The Boreal-Arctic Wetland and Lake Dataset (BAWLD)

David Olefeldt[1], Mikael Hovemyr[2], McKenzie A. Kuhn[1], David Bastviken[3], Theodore J. Bohn[4], John Connolly[5], Patrick Crill[6], Eugénie S. Euskirchen[7,8], Sarah A. Finkelstein[9], Hélène Genet[8], Guido Grosse[10,11], Lorna I. Harris[1], Liam Heffernan[12], Manuel Helbig[13], Gustaf Hugelius[2,14], Ryan Hutchins[15], Sari Juutinen[16], Mark J. Lara[17,18], Avni Malhotra[19], Kristen Manies[20], A. David McGuire[8], Susan M. Natali[21], Jonathan A. O'Donnell[22], Frans-Jan W. Parmentier[23,24], Aleksi Räsänen[25], Christina Schädel[26], Oliver Sonnentag[27], Maria Strack[28], Suzanne E. Tank[29], Claire Treat[10], Ruth K. Varner[2,30], Tarmo Virtanen[25], Rebecca K. Warren[31], Jennifer D. Watts[21]

[1] Department of Renewable Resources, University of Alberta, Edmonton, AB, T6G 2G7, Canada
[2] Department of Physical Geography, Stockholm University, 10691 Stockholm, Sweden
[3] Department of Thematic Studies – Environmental Change, Linköping University, 58183 Linköping, Sweden
[4] WattIQ, 400 Oyster Point Blvd. Suite 414, South San Francisco, CA, 94080, USA
[5] Department of Geography, School of Natural Sciences, Trinity College Dublin, Dublin 2, Ireland
[6] Department of Geological Sciences, Stockholm University, 10691 Stockholm, Sweden
[7] Department of Biology and Wildlife, University of Alaska Fairbanks, Fairbanks, AK 99775, USA
[8] Institute of Arctic Biology, University of Alaska Fairbanks, Fairbanks, AK 99775, USA
[9] Department of Earth Sciences, University of Toronto, Toronto, ON, M5S 3B1, Canada
[10] Alfred Wegener Institute, Helmholtz Centre for Polar and Marine Research, Permafrost Research Section, 14473 Potsdam, Germany
[11] Institute of Geosciences, University of Potsdam, 14476 Potsdam, Germany
[12] Department of Ecology and Genetics, Uppsala University, 752 36 Uppsala, Sweden
[13] Department of Physics and Atmospheric Science, Dalhousie University, Halifax, NS, B3H 4R2, Canada
[14] Bolin Centre for Climate Research, Stockholm University, 10691 Stockholm, Sweden
[15] Department of Earth and Environmental Sciences, University of Waterloo, Waterloo, ON, N2L 3G1, Canada
[16] Ecosystems and Environment Research Program, University of Helsinki, FI-00014 Helsinki, Finland
[17] Department of Plant Biology, University of Illinois, Urbana, IL 61801, USA
[18] Department of Geography, University of Illinois, Urbana, IL 61801, USA
[19] Department of Earth System Science, Stanford University, Stanford, CA 94305, USA
[20] U.S. Geological Survey, Menlo Park, CA, USA
[21] Woodwell Climate Research Center, Falmouth, MA 02540, USA
[22] Arctic Network, National Park Service, Anchorage, AK 99501 USA
[23] Centre for Biogeochemistry in the Anthropocene, Department of Geosciences, University of Oslo, 0315 Oslo, Norway
[24] Department of Physical Geography and Ecosystem Science, Lund University, 223 62 Lund, Sweden
[25] Ecosystems and Environment Research Programme, Faculty of Biological and Environmental Sciences, 00014 University of Helsinki, Finland
[26] Center for Ecosystem Science and Society, Northern Arizona University, Flagstaff, AZ 86011, USA
[27] Département de Géographie, Université de Montréal, Montréal, QC, Canada
[28] Department of Geography and Environmental Management, University of Waterloo, Waterloo, ON, N2L 3G1, Canada
[29] Department of Biological Sciences, University of Alberta, Edmonton, AB, T6G 2E9, Canada
[30] Department of Earth Sciences and Institute for the Study of Earth, Oceans and Space, University of New Hampshire, Durhan, NH 03824, USA
[31] National Boreal Program, Ducks Unlimited Canada, Edmonton, AB, T5S 0A2, Canada

*Correspondence to*: David Olefeldt (olefeldt@ualberta.ca)



**Abstract.**

Methane emissions from boreal and arctic wetlands, lakes, and rivers are expected to increase in response to warming and associated permafrost thaw. However, the lack of appropriate land cover datasets for scaling field-measured methane emissions to circumpolar scales has contributed to a large uncertainty for our understanding of present-day and future methane emissions. Here we present the Boreal-Arctic Wetland and Lake Dataset (BAWLD), a land cover dataset based on an expert assessment, extrapolated using random forest modelling from available spatial datasets of climate, topography, soils, permafrost conditions, vegetation, wetlands, and surface water extents and dynamics. In BAWLD, we estimate the fractional coverage of five wetland, seven lake, and three river classes within $0.5 \times 0.5°$ grid cells that cover the northern boreal and tundra biomes (17% of the global land surface). Land cover classes were defined using criteria that ensured distinct methane emissions among classes, as indicated by a co-developed comprehensive dataset of methane flux observations. In BAWLD, wetlands occupied $3.2 \times 10^6$ km$^2$ (14% of domain) with a 95% confidence interval between 2.8 and $3.8 \times 10^6$ km$^2$. Bog, fen, and permafrost bog were the most abundant wetland classes, covering ~28% each of the total wetland area, while the highest methane emitting marsh and tundra wetland classes occupied 5 and 12%, respectively. Lakes, defined to include all lentic open-water ecosystems regardless of size, covered $1.4 \times 10^6$ km$^2$ (6% of domain). Low methane-emitting large lakes (>10 km$^2$) and glacial lakes jointly represented 78% of the total lake area, while high-emitting peatland and yedoma lakes covered 18 and 4%, respectively. Small (<0.1 km$^2$) glacial, peatland, and yedoma lakes combined covered 17% of the total lake area, but contributed disproportionally to the overall spatial uncertainty of lake area with a 95% confidence interval between 0.15 and $0.38 \times 10^6$ km$^2$. Rivers and streams were estimated to cover $0.12 \times 10^6$ km$^2$ (0.5% of domain) of which 8% was associated with high-methane emitting headwaters that drain organic-rich landscapes. Distinct combinations of spatially co-occurring wetland and lake classes were identified across the BAWLD domain, allowing for the mapping of "wetscapes" that will have characteristic methane emission magnitudes and sensitivities to climate change at regional scales. With BAWLD, we provide a dataset which avoids double-accounting of wetland, lake and river extents, and which includes confidence intervals for each land cover class. As such, BAWLD will be suitable for many hydrological and biogeochemical modelling and upscaling efforts for the northern Boreal and Arctic region, in particular those aimed at improving assessments of current and future methane emissions. Data is freely available at https://doi.org/10.18739/A2C824F9X (Olefeldt et al., 2021).



## 70    1 Introduction

Emissions of methane ($CH_4$) from abundant wetlands, lakes, and rivers located in boreal and arctic regions are expected to substantially increase this century due to rapid climate warming and associated permafrost thaw (Walter Anthony et al., 2018; Ito, 2019; Hugelius et al., 2020; Schneider von Deimling et al., 2015; Zhang et al., 2017). However, predicting future $CH_4$ emissions is highly uncertain, as estimates of present-day $CH_4$ emissions from boreal and arctic regions are poorly

constrained, ranging between 21 and 77 Tg $CH_4$ $yr^{-1}$ (Saunois et al., 2020; Peltola et al., 2019; Wik et al., 2016; Treat et al., 2018; McGuire et al., 2012; Watts et al., 2014; Thompson et al., 2018; Zhu et al., 2015; Tan et al., 2016; Walter Anthony et al., 2016). Estimates of high-latitude $CH_4$ emissions vary between approaches, with generally lower estimates from atmospheric inversions (top-down estimates), than from field-measured $CH_4$ emissions data paired with land cover data (bottom-up estimates) (Saunois et al., 2020; McGuire et al., 2012). Low accuracy of high-latitude land cover datasets for

wetland and lake distributions, and their classification, represent key sources of uncertainty for estimates of high-latitude $CH_4$ emissions and may contribute to the discrepancies between bottom-up and top-down estimates. A limitation of many currently available land cover datasets is an insufficient differentiation between wetland, lake, and river classes that are known to have distinct $CH_4$ emissions (Bruhwiler et al., 2021; Bohn et al., 2015; Marushchak et al., 2016; Melton et al., 2013).


There are several challenges when using remote sensing approaches to map distinct wetland, lake, and river classes at the circumpolar scale. Many small or narrow wetland ecosystems with high methane $CH_4$ emissions are located along lake shorelines, along stream networks, or in polygonal tundra terrain, and are thus difficult to map as image resolution can be inadequate (Wickland et al., 2020; Cooley et al., 2017; Virtanen and Ek, 2014; Liljedahl et al., 2016). Wetland detection can

further be complicated by the presence of tree species in wetlands, e.g. Scots pine (*Pinus sylvestris*), black spruce (*Picea mariana*), and tamarack (*Larix laricina*), that are also found in non-wetland boreal forests, making differentiation of treed wetlands from non-wetland forests difficult. Using spectral signatures to differentiate and map distinct wetland classes can further be difficult due to seasonal variation in inundation or phenology, poor differentiation between ecosystems (e.g. similarities between different peatland classes), or high spectral diversity within classes due to shifts in vegetation along

subtle environmental gradients (Räsänen and Virtanen, 2019; Vitt and Chee, 1990; Chasmer et al., 2020). Vegetation composition and spectral signatures of wetland classes can also vary between different high-latitude regions, e.g. with shifts in dominant tree and shrub species between North America and Eurasia (Raynolds et al., 2019), and be influenced for decades by wildfires (Chen et al., 2021; Helbig et al., 2016). Active microwave remote sensing can help detect inundated wetlands and saturated soils, but has limitations due to its computational requirements, coarse resolution, and issues with

detecting rarely inundated peatlands (Beck et al., 2021; Duncan et al., 2020). Accurate mapping of wetlands that include differentiation among distinct wetland classes requires substantial ground truthing, something which has only been done consistently at local and regional scales (Terentieva et al., 2016; Chasmer et al., 2020; Bryn et al., 2018; Lara et al., 2018;





Canadian Wetland Inventory Technical Committee, 2016). Similar issues arise for lakes, rivers, and streams. While larger lakes and rivers have been mapped with high precision (Messager et al., 2016; Linke et al., 2019), the highest $CH_4$ emissions

are generally from ponds, pools, and low-order streams that are too small to be accurately detected by anything other than very high-resolution imagery (Muster et al., 2017). Statistical approaches are often used to model the distribution and abundance of small open-water ecosystems, yielding large uncertainties (Holgerson and Raymond, 2016; Cael and Seekell, 2016; Muster et al., 2019). Remote sensing approaches are also inadequate in assessing other key variables known to influence lake $CH_4$ emissions, including lake genesis, depth, and sediment characteristics (Messager et al., 2016; Brosius et

al., 2021; Smith et al., 2007; Lara et al., 2021). Another key issue is that wetlands and lakes often are mapped separately, allowing for potential double-counting of ecosystems in both wetland and lake inventories (Thornton et al., 2016; Saunois et al., 2020).

Emissions of $CH_4$ from boreal and arctic ecosystems range from uptake to some of the highest emissions observed globally

(Turetsky et al., 2014; Knox et al., 2019; Glagolev et al., 2011; St Pierre et al., 2019). Net ecosystem $CH_4$ emissions are a balance between microbial $CH_4$ production (methanogenesis) and oxidation (methanotrophy), a balance further influenced by the dominant transport pathway; diffusion, ebullition, and plant-mediated transport (Bridgham et al., 2013; Bastviken et al., 2004). For wetlands, defined as ecosystems with temporally or permanently saturated soils and biota adapted to anoxic conditions, $CH_4$ emissions in boreal and arctic regions are primarily influenced by water table position, soil temperatures,

and vegetation composition and productivity (Olefeldt et al., 2013; Treat et al., 2018). Marshes and tundra wetlands are characterized by frequent or permanent inundation and dominant graminoid vegetation that enhance methanogenesis and facilitates plant-mediated transport, and thus generally have high $CH_4$ emissions (Knoblauch et al., 2015; Juutinen et al., 2003). Conversely, peat-forming bogs and fens generally have a water table at or below the soil surface, less graminoid vegetation and instead vegetation dominated by mosses, lichens, and shrubs, resulting in typically low to moderate $CH_4$

emissions (Bubier et al., 1995; Pelletier et al., 2007). Permafrost conditions in peatlands can cause the surface to be elevated and dry, with cold soil conditions where methanogenesis is inhibited, leading to low $CH_4$ emissions or even uptake (Bäckstrand et al., 2008; Glagolev et al., 2011). Non-wetland boreal forests and tundra ecosystems generally have net $CH_4$ uptake, as methanotrophy outweighs any methanogenesis (Lau et al., 2015; Juncher Jørgensen et al., 2015; Whalen et al., 1992). The transition from terrestrial to aquatic ecosystems is not always well defined, and several wetland classification

systems consider shallow, open-water ecosystems as a distinct wetland class (Rubec, 2018). The transition from vegetated to open water ecosystems is however associated with shifts in apparent primary controls of $CH_4$ emissions, including a shift towards increased importance of ebullition (Bastviken et al., 2004). For lakes, when defined to include all lentic open-water ecosystems regardless of size (e.g. including peatland ponds), spatial variability in $CH_4$ emissions is primarily linked to water depth and the quantity and origin of the organic matter of the sediment (Heslop et al., 2020; Li et al., 2020). As such,

lake $CH_4$ emissions are generally higher for smaller lakes and for lakes with organic-rich sediments (Wik et al., 2016; Holgerson and Raymond, 2016), which are extremely abundant in many high-latitude regions (Muster et al., 2017). The $CH_4$





emitted from streams and rivers is largely derived from the soils that are drained, and as such emissions generally are higher in smaller streams draining wetland-rich watersheds (Wallin et al., 2018; Stanley et al., 2016). It is overall likely that studies of $CH_4$ emissions from boreal and arctic ecosystems have focused disproportionally on sites with higher $CH_4$ emissions

(Olefeldt et al. 2013). A focus on high-emitting sites is warranted for understanding site-level controls on $CH_4$ emissions, but may potentially cause bias of bottom-up $CH_4$ scaling approaches if they lack appropriate differentiation between various wetland and lakes classes in the used land cover datasets.

There is currently no spatial dataset available that has information on the distribution and abundance of wetland, lake, and

river classes defined specifically for the purpose of estimating boreal and arctic $CH_4$ emissions. However, a large number of spatial datasets have partial, but relevant, information. This includes circumpolar spatial data of soil types (Hugelius et al., 2013; Strauss et al., 2017), vegetation (Olson et al., 2001; Walker et al., 2005), surface water extent and dynamics (Pekel et al., 2016), lake sizes and numbers (Messager et al., 2016), topography (Gruber, 2012), climate (Fick and Hijmans, 2017), permafrost conditions (Gruber, 2012; Brown et al., 2002), river networks (Linke et al., 2019), and previous estimates of total

wetland cover (Matthews and Fung, 1987; Bartholomé and Belward, 2005). By integrating quantitative spatial data with expert knowledge it is possible to model new spatial data for specific purposes (Olefeldt et al., 2016). Researchers with interests in the boreal and arctic have considerable knowledge of the presence and relative abundance of typical wetland and lake classes in various high-latitude regions, along with the ability for satellite image interpretation and the judgement to define parsimonious land cover classes suitable for $CH_4$ scaling.


Here we present the Boreal-Arctic Wetland and Lake Dataset (BAWLD), an expert knowledge-based land cover dataset. Developed in concert with a comprehensive dataset of observed $CH_4$ fluxes from high-latitude aquatic ecosystems (Kuhn et al., 2021), BAWLD includes five wetland, seven lake, and three river classes with distinct $CH_4$ emissions. Coverage of each wetland, lake, and river class within 0.5° grid cells was modelled through random forest regressions based on expert

assessment data and available relevant spatial data. The approach aims to reduce issues with bias in representativeness of empirical data, to reduce issues of overlaps in wetland and lake extents, and to allow for the partitioning of uncertainty of $CH_4$ emissions estimates to $CH_4$ emission magnitudes or areal extents of different land cover classes. As such, BAWLD will facilitate improved bottom-up estimates of high-latitude $CH_4$ emissions, and will be suitable for use in process-based models and as an a-priori input to inverse modelling approaches. The land cover dataset will be suitable for further uses, especially

for questions related to high-latitude hydrology and biogeochemistry. Lastly, BAWLD allows for the definition of "wetscapes"; regions with distinct co-occurrences of specific wetland and lake classes, and which thus can be used to understand regional responses to climate change and as a way to visualize the landscape diversity of the boreal and arctic domain.



**Table 1. Description of data sources and layers extracted into the BAWLD 0.5° grid cell network.**

| Dataset and extracted layers | Dataset and extracted layers |
|---|---|
| WorldClim V2 (Fick and Hijmans, 2017) | Reference information |
| - *WC2-MAAT: Mean annual average air temperature 1970-2000. (˚C)* | - *LAT: Latitude (˚)* |
| - *WC2-MAAP: Mean annual average precipitation 1970-2000 (mm)* | - *LONG: Longitude (˚)* |
| - *WC2-CMI: Climate moisture index 1970-2000 (mm)* | - *SHORE: Coastal shoreline presence in cell (yes/no)* |
| | |
| Circum-Arctic Map of Permafrost and Ground-Ice (Brown et al., 2002) | Northern Circumpolar Soil Carbon Dataset (Hugelius et al., 2014) |
| - *CAPG-CON: Continuous permafrost. (%)* | - *NCS-HSO: Histosol soils; non-permafrost organic soils. (%)* |
| - *CAPG-DIS: Discontinuous permafrost. (%)* | - *NCS-HSE Histel soils: permafrost organic soils. (%)* |
| - *CAPG-SPO: Sporadic permafrost. (%)* | - *NCS-AQU: Aqueous soils: non-organic wetland soils. (%)* |
| - *CAPG-ISO: Isolated permafrost. (%)* | - *NCS-ROC: Rocklands. (%)* |
| - *CAPG-XHF: Land with thick overburden and >20% ground-ice. (%)* | - *NCS-GLA: Glaciers. (%)* |
| - *CAPG-XMF: Land with thick overburden and 10-20% ground-ice. (%)* | - *NCS-H2O: Open water. (%)* |
| - *CAPG-XLF: Land with thick overburden and <10% ground-ice. (%)* | |
| - *CAPG-XHR: Land with thin overburden and >10% ground-ice. (%).* | |
| - *CAPG-XLR: Land with thin overburden and <10% ground-ice. (%)* | |
| - *CAPG-REL: Land with relict permafrost. (%)* | |
| | |
| BasinATLAS (Linke et al., 2019) | Global Lakes and Wetland Dataset (Lehner and Döll, 2004) |
| - *BAS-RIV: River area. (%)* | - *GLWD-RIV: Rivers, 6th order rivers or greater. (%)* |
| | |
| Circumpolar Arctic Vegetation Map (CAVM Team 2003) | Terrestrial Ecoregions of the World (Olson et al., 2001) |
| - *CAVM-BAR: Barren Tundra. (%)* | - *TEW-BOR: Fractional cover of boreal ecoregion/ (%)* |
| - *CAVM-GRA: Graminoid Tundra. (%)* | - *TEW-TUN: Fractional cover of tundra ecoregion. (%)* |
| - *CAVM-SHR: Shrubby Tundra. (%)* | - *TEW-GLA: Fractional cover of glaciers. (%)* |
| - *CAVM-WET: Wet Tundra. (%)* | |
| | |
| HydroLakes (Messager et al., 2016) | Global Land Cover Database 2000 (Bartholomé and Belward, 2005) |
| - *HL-LAR: Lakes >10 km2. (%)* | - *GLC2-H2O: Water Bodies, natural and artificial. (%)* |
| - *HL-MID: Lakes between 10 km2 and 0.1 km2. (%)* | - *GLC2-RFSM: Regularly Flooded Shrub and/or Herbaceous Cover. (%)* |
| - *HL-SHO: Shoreline density (length/area) of lakes >0.1 km2. (m/m2)* | - *GLC2-FOR: Forest cover. (%)* |
| | |
| Global Inundation Map (Fluet-Chouinard et al., 2015) | Dataset of Ice-Rich Yedoma Permafrost (Strauss et al., 2017) |
| - *GIM-MAMI: Mean annual minimum inundation. (%)* | - *IRYP-YED: Yedoma ground. (%)* |
| - *GIM-MAMA: Mean annual maximum inundation. (%)* | |
| | |
| GlobLand30 (Chen et al., 2015) | Permafrost zonation and Terrain Ruggedness Index (Gruber, 2012) |
| - *GL30-H2O: Water bodies: including lakes, rivers, reservoirs. (%)* | - *PZI-PERM: Permafrost ground. (%)* |
| - *GL30-WET: Wetlands: marshes, floodplains, shrub wetland, peatlands. (%)* | - *PZI-FLAT: Flat topography. (%)* |
| - *GL30-TUN: Tundra: shrub, herbaceous, wet, and barren tundra. (%)* | - *PZI-UND: Undulating topography. (%)* |
| - *GL30-ART: Artificial Surfaces: cities, industry, transport. (%)* | - *PZI-HILL: Hilly topography. (%)* |
| - *GL30-ICE: Permanent snow and ice. (%)* | - *PZI-MTN: Mountainous topography. (%)* |
| | - *PZI-RUG: Rugged topography. (%)* |
| | |
| Global Surface Water (Pekel et al., 2016) | Global Wetlands (Matthews and Fung, 1987) |
| - *GSW-RAR: Rarely inundated; open water in 0 to 5% of occasions. (%)* | -*GWET-IN: Inundation and presence of wetlands. (%).* |
| - *GSW-OCC: Occasionally inundated; open water 5 to 50%. (%)* | |
| - *GSW-REG: Regularly inundated; open water 50 to 95%. (%)* | |
| - *GSW-PER: Permanent open water; open water 95 to 100%. (%)* | |



## 2 Development of the Boreal-Arctic Wetland and Lake Dataset

### 2.1 Study domain and harmonization of available spatial data

The BAWLD domain includes all of the northern boreal and tundra ecoregions, and also areas of rock and ice at latitudes
>50°N (Olson et al., 2001). The BAWLD domain thus covers $25.5 \times 10^6$ km$^2$, or 17% of the global land surface. Although northern peat-forming wetlands can also be found in temperate ecoregions, our decision to define the southern limit of BAWLD by the transition from boreal to temperate ecoregions was based on the greater human footprint and the increased biogeographic diversity of temperate ecoregions, which would require additional land cover classes (Venter et al., 2016). A network of 0.5° grid cells, cropped along coasts and at the transition from boreal to temperate ecoregions, was created for the
BAWLD domain.

Grid cells in BAWLD were populated with data from 15 publicly available spatial datasets, yielding 53 variables with spatial information (Table 1). Most datasets that were included have data at higher resolution than the 0.5° BAWLD grid cells, hence information was averaged for each grid cell. For datasets where the spatial resolution was coarser or where spatial data were not aligned with the 0.5° grid cells, data was first apportioned into BAWLD grid cells before area-weighted averages
were calculated. Climate data from the WorldClim2 (WC2) dataset (Fick and Hijmans, 2017) were averaged for each grid cell, including "mean annual air temperature", "mean annual precipitation", and "climate moisture index". Information on soils and permafrost conditions were summarized as fractional coverage within each grid cell, and included "permafrost extent" from the Permafrost Zonation and Terrain Ruggedness Index (PZI) dataset (Gruber, 2012), permafrost zonation,
ground ice content, and overburden thickness from the Circum-Arctic Map of Permafrost and Ground-Ice (CAPG) dataset (Brown et al., 2002), "yedoma ground" from the Ice-Rich Yedoma Permafrost (IRYP) dataset (Strauss et al., 2017), and non-permafrost peat "histosol", permafrost peat "histel", and "aqueous" wetland soils from the Northern Circumpolar Soil Carbon Database (NCSCD; hereafter NCS) (Hugelius et al., 2013). Four independent datasets provided information on wetland coverage, although without further differentiation between distinct wetland classes; the "regularly flooded shrub
and/or herbaceous cover" area from the Global Land Cover Database 2000 (GLC2) (Bartholomé and Belward, 2005), the "wetlands" area in the GlobLand30 (GL30) dataset (Chen et al., 2015), and the "inundation and presence of wetlands" area from the Global Wetlands (GWET) dataset (Matthews and Fung, 1987), and the Circumpolar Arctic Vegetation Map (CAVM) dataset (Walker et al., 2005). Two datasets provided information of the extent of forested regions; the GLC2 and the Terrestrial Ecoregions of the World (TEW) dataset (Olson et al., 2001), while three datasets provided information on the
extents of tundra vegetation; the CAVM, the GL30, and the TEW. Three datasets provided information on extent of glaciers and permanent snow; the NCS, the GL30, and the TEW. The NCS dataset also provided information about the extents of "rocklands", while the PZI dataset had extents of topographic ruggedness ("flat", "undulating", "hilly", "mountainous", and "rugged"). Information on river extents was found in two datasets; the "river area" in the BasinATLAS (BAS) dataset (Linke et al., 2019), and "rivers" in the Global Lakes and Wetland (GLW) dataset, which includes 6[th] order rivers and greater



(Lehner and Döll, 2004). Inundation dynamics was provided by two datasets, with "mean annual minimum" and "mean annual maximum" inundation in the Global Inundation Map (GIM) dataset (Fluet-Chouinard et al., 2015), and an analysis of temporal inundation from the Global Surface Water (GSW) dataset (Pekel et al., 2016) where we defined inundation of individual 30 m pixels as being inundated "rarely" (>0 to 5% of all available Landsat images), "occasionally" (5 to 50%), "regularly" (50 to 95%), or "permanently" (95 to 100%). Four datasets included information about static extents of open

water, including "open water" in NCS, "water bodies" in GL30, "water bodies" in GLC2, and information about lakes in the Hydrolakes (HL) dataset (Messager et al., 2016), where we differentiated between the area of "large lakes" (lakes >10 km$^2$), and "midsize lakes" (lakes between 0.1 and 10 km$^2$).

## 2.2 Land cover classes in BAWLD

The land cover classification in BAWLD was constructed with the goal to enable upscaling of CH$_4$ fluxes for large spatial

extents. As such, we aimed to include as few classes as possible to facilitate for large-scale mapping, while still including classes that allow for separation among ecosystems with distinct hydrology, ecology, biogeochemistry and thus net CH$_4$ fluxes. The BAWLD land cover classification is hierarchical; with five wetland classes, seven lake classes, and three river classes, along with four other classes; glaciers, dry tundra, boreal forest, and rocklands. The class descriptions below were provided to all experts for their land cover assessments, and thus effectively serve as the BAWLD class definitions.

### 2.2.1 Wetland Classes

Wetlands are defined by having a water table near or above the land surface for sufficient time to cause the development of wetland soils (either mineral soils with redoximorphic features, or organic soils with > 40 cm peat), and the presence of plant species with adaptations to wet environments (Hugelius et al., 2020; Canada Committee on Ecological (Biophysical) Land Classification et al., 1997; Jorgenson et al., 2001). Wetland classifications for boreal and arctic biomes can focus either on

small-scale wetland classes that have distinct hydrological regimes, vegetation composition, and biogeochemistry, or on larger-scale wetland complexes that are comprised of distinct patterns of smaller wetland and open-water classes (Gunnarsson et al., 2014; Terentieva et al., 2016; Masing et al., 2010; Glaser et al., 2004). While larger-scale wetland complexes are easier to identify through remote sensing techniques (e.g. patterned fens comprised of higher elevation ridges and inundated hollows), our classification focuses on wetland classes due to greater homogeneity of hydrological, ecological,

and biogeochemical characteristics that regulate CH$_4$ fluxes (Heiskanen et al., 2021).

Several boreal countries identify four main wetland classes, differentiated primarily based on hydrodynamic characterization; bogs, fens, marshes, and swamps (Gunnarsson et al., 2014; Canada Committee on Ecological (Biophysical) Land Classification et al., 1997; Masing et al., 2010). The BAWLD classification follows this general framework, but further

uses the presence or absence of permafrost as a primary characteristic for classification and excludes a distinct swamp class, yielding five classes; *Bogs*, *Fens*, *Marshes*, *Permafrost Bogs*, and *Tundra Wetlands* (Figure 1). The swamp class was omitted

due to the wide range of moisture and nutrient conditions of swamps, as well as the limited number of studies of swamp $CH_4$

fluxes (Kuhn et al., 2021). We instead included swamp ecosystems in expanded descriptions of *Bogs*, *Fens*, and *Marshes*.

The presence or absence of near-surface permafrost was used as a primary characteristic to distinguish between *Permafrost*

*Bogs* and *Bogs,* and to distinguish *Tundra Wetlands* from *Marshes* and *Fens*. The presence or absence of near-surface

permafrost is considered key for controlling $CH_4$ emissions given its influence on hydrology, and for the potential of

permafrost thaw and thermokarst collapse to cause rapid non-linear shifts to $CH_4$ emissions (Bubier et al., 1995; Turetsky et

al., 2002; Malhotra and Roulet, 2015). Finally, while some classifications include shallow (e.g. 2 m depth), open-water

ecosystems within the definition of wetlands (Gunnarsson et al., 2014; Canada Committee on Ecological (Biophysical) Land

Classification et al., 1997), we have included all open-water ecosystems without emergent vegetation within the lake classes

(see below) due to the strong influence of emergent vegetation in controlling $CH_4$ emissions (Juutinen et al., 2003).

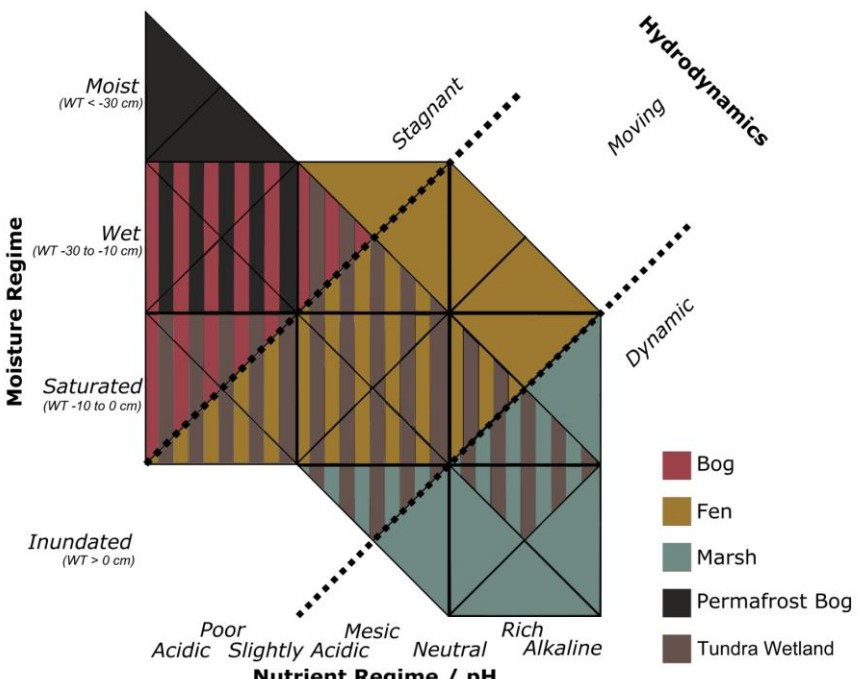

**Figure 1. Descriptions of wetland classes in BAWLD as distinguished based on the moisture regime, the nutrient/pH regime,**
**hydrodynamics, and the presence/absence of permafrost.**

*Bogs* are described as ombrotrophic peatland ecosystems, i.e. only dependent on precipitation, and snowmelt for water

inputs. Peat thickness is at least 40 cm, with maximum thickness > 10 m. The peat profile is not affected by permafrost,

although in some climatically colder settings there may be permafrost below the peat profile. *Bogs* are wet to saturated

ecosystems, often with small-scale (<10 m) microtopographic variability, with stagnant water and a water table that rarely is

above the surface or more than 50 cm below the surface (Figure 1). *Bogs* have low pH (<5), low concentrations of dissolved



ions, and low nutrient availability resulting from a lack of hydrological connectivity to surrounding mineral soils. Vegetation is commonly dominated by *Sphagnum* mosses, lichens, and woody shrubs, and can be either treed or treeless (Beaulne et al., 2021). Our description of *Bogs* also includes what is commonly classified as treed swamps, which generally represent

ecotonal transitions between peatlands and upland forests (Canada Committee on Ecological (Biophysical) Land Classification et al., 1997). Emissions of $CH_4$ from *Bogs* are low to moderate in comparison with other classes (Kuhn et al., 2021).

*Fens* are described as minerotrophic peatland ecosystems, i.e. hydrologically connected to surrounding mineral soils through

surface water or groundwater inputs. A *Fen* peat profile is at least 40 cm thick (Gorham et al. 1991), although maximum thickness is generally less than for bogs. The peat profile is not affected by permafrost. *Fens* are wet to saturated ecosystems, with generally slow-moving water (Figure 1). *Fens* have widely ranging nutrient regimes and levels of dissolved ions depending on the degree and type of hydrological connectivity to their surroundings, ranging from poor fens to rich fens. Vegetation largely depends on wetness and nutrient availability, where more nutrient poor fens can have *Sphagnum* mosses,

shrubs, and trees, while rich fens are dominated by brown mosses, graminoids (sedges, rushes), herbaceous plants, and sometimes coniferous or deciduous trees (e.g. willows, birch, larch). Our description of *Fens* also includes what is commonly classified as shrubby swamps, which often are associated with riparian ecotones and lake shorelines. *Fen* $CH_4$ emissions are moderate to high in comparison to other classes (Kuhn et al., 2021).

*Marshes* are minerotrophic wetlands with dynamic hydrology, and often high nutrient availability (Figure 1). Vegetation is dominated by emergent macrophytes, including tall graminoids such as rushes, reeds, grasses and sedges – some of which can persist in settings with >1.5 m of standing water. *Marshes* are saturated to inundated wetlands, often with highly fluctuating water levels as they generally are located along shorelines of lake or coasts, along streams and rivers, or on floodplains and deltas. It is common for marshes to exhibit both flooded and dry periods. Dry periods facilitate

decomposition of organic matter, and can prevent the build-up of peat. As such *Marshes* generally have mineral soils, although some settings allow for the accumulation of highly humified organic layers – sometimes indicating ongoing succession towards a peatland ecosystem. Salinity can vary depending on water sources, with brackish to saline conditions in some areas of groundwater discharge, or in coastal settings. While highly saline marshes can have low $CH_4$ emissions due to inhibition of methanogenesis, *Marshes* generally have high to very high $CH_4$ emissions compared to other classes (Kuhn et

al., 2021).

*Permafrost Bogs* are peatland ecosystems, although the peat thickness in cold climates is often relatively shallow. *Permafrost Bogs* have a seasonally thawed active layer that is 30 to 70 cm thick, with the remainder of the peat profile perennially frozen (i.e. permafrost). Excess ground-ice and ice expansion often elevate *Permafrost Bogs* up to a few meters

above their surroundings, and as such they are ombrotrophic and generally the wetland class with the driest soils (Figure 1).





*Permafrost Bogs* have moist to wet soil conditions, often with a water table that follows the base of the seasonally developing thawed soil layer. Ombrotrophic conditions cause nutrient-poor conditions, and the vegetation is dominated by lichens, *Sphagnum* mosses, woody shrubs, and sometimes stunted coniferous trees. *Permafrost Bogs* are often interspersed in a fine-scale mosaic (10 to 100 m) with other wetland classes, e.g. *Bogs* and *Fens*. Common *Permafrost Bog* landforms

include palsas, peat plateaus, and the elevated portions of high- and low-center polygonal peatlands. *Permafrost Bogs* have very low $CH_4$ emissions, and sometimes even $CH_4$ uptake (Kuhn et al., 2021).

*Tundra Wetlands* are treeless ecosystems with saturated to inundated conditions, most commonly with near surface permafrost (Figure 1). *Tundra Wetlands* can have either mineral soils or shallow organic soils, and generally receive surface

or near-surface waters from their surroundings, as permafrost conditions preclude connectivity to deeper groundwater sources. Vegetation is dominated by short emergent vegetation, including sedges and grasses, with mosses and shrubs in slightly drier sites. *Tundra Wetlands* have lower maximum depth of standing water than *Marshes,* due to the shorter vegetation. *Tundra Wetlands* can be found in basin depressions, in low-center polygonal wetlands, and along rivers, deltas, lake shorelines, and on floodplains in regions of continuous permafrost. Despite the name, limited wetlands with these

characteristics (hydrology, permafrost conditions, and vegetation) can also be found within the continuous permafrost zone in boreal and sub-arctic regions (Virtanen et al., 2016). *Tundra Wetlands* have moderate to high $CH_4$ emissions compared to other classes (Kuhn et al., 2021).

### 2.2.2 Lake Classes

Lakes in BAWLD are considered to include all lentic open-water ecosystems, regardless of surface area and depth of

standing water. It is common in ice-rich permafrost lowlands and peatlands for open-water bodies to have shallow depths, often less than two meters, even when surface areas are up to hundreds of $km^2$ in size (Grosse et al., 2013). While small, shallow open-water bodies often are included in definitions of wetlands (Canada Committee on Ecological (Biophysical) Land Classification et al., 1997; Gunnarsson et al., 2014; Treat et al., 2018), we include them here within the lake classes as controls on net $CH_4$ emissions depend strongly on the presence or absence of emergent macrophytes (Juutinen et al., 2003).

Further classification of lakes in BAWLD is based lake size and lake genesis where lake genesis influences lake bathymetry and sediment characteristics. Previous global spatial inventories of lakes include detailed information on size and location of individual larger lakes (Messager et al., 2016; Downing et al., 2012), but do not include open-water ecosystems <0.1 $km^2$ in size, and do not differentiate between lakes of different genesis (e.g. tectonic, glacial, organic, and yedoma lakes). Small water bodies are disproportionately abundant in some high latitude environments (Muster et al., 2019), have high emissions

of $CH_4$ (Holgerson and Raymond, 2016), and therefore require explicit classification apart from larger water-bodies. Furthermore, lake genesis and sediment type haven been shown to influence net $CH_4$ flux from lakes (Wik et al., 2016). In BAWLD we thus differentiate between large (>10 $km^2$), midsize (0.1 to 10 $km^2$) and small (<0.1 $km^2$) lake classes, and further differentiate between three lake types for midsize and small lakes; peatland, yedoma, and glacial lakes.



*Small* and *Midsize Peatland Lakes* are described as lakes with thick organic sediments that are mainly found adjacent to or surrounded by peatlands, or in lowland tundra regions with organic-rich soils. *Small Peatland Lakes* includes the numerous small pools often found in extensive peatlands and lowland tundra regions, e.g. including the open-water parts of string fens and polygonal peatlands. *Peatland Lakes* generally form as a result of interactions between local hydrology and the accumulation of peat which can create open water pools and lakes (Garneau et al., 2018; Harris et al., 2020), but can also

form in peatlands as a result of permafrost dynamics (Liljedahl et al., 2016; Sannel and Kuhry, 2011). As such, these lakes with thick organic sediments are often shallow and have a relatively low shoreline development index. *Peatland lakes* have dark waters with high concentrations of dissolved organic carbon. *Small Peatland Lakes* generally have higher $CH_4$ emissions than *Midsize Peatland Lakes*, but observations from both classes range from moderate to very high $CH_4$ emissions, with roughly half of emissions attributed to ebullition (Kuhn et al., 2021).


*Small* and *Midsize Yedoma Lake*s are exclusive to non-glaciated regions of eastern Siberia, Alaska, and the Yukon where yedoma deposits accumulated during the Pleistocene (Strauss et al., 2017). Yedoma permafrost soils are ice-rich and contain fine-grained, organic-rich loess which was deposited by wind and accumulated upwards in parallel with permafrost aggradation, thus limiting decomposition and facilitating organic matter burial (Schirrmeister et al., 2013). Notable

thermokarst features, including lakes, often develop when yedoma permafrost thaws, causing labile organic matter to become available for microbial mineralization (Walter Anthony et al., 2016). *Small Yedoma lakes* typically represent younger thermokarst features, whereas *Midsized Yedoma Lakes* represent later stages of thermokarst lake development. *Small Yedoma Lakes* are thus more likely to have actively thawing and expanding lake edges where $CH_4$ emissions can be extremely high, largely driven by high ebullition emissions (Walter Anthony et al., 2016). Century-scale development of

yedoma lakes can shift the main source of $CH_4$ production from yedoma deposits to new organic-rich sediment that accumulate from allochtonous and autochthonous sources – resulting in such lakes here being considered as *Peatland Lakes*. *Midsize Yedoma Lakes* have lower $CH_4$ emissions than *Midsize Peatland Lakes*, while *Small Yedoma Lakes* have similar $CH_4$ emissions as *Small Peatland Lakes* – albeit with a greater proportion attributed to ebullition (Kuhn et al., 2021).

*Small* and *Midsize Glacial Lakes* include all lakes with organic-poor sediments – predominately those formed through glacial or post-glacial processes, e.g. kettle lakes and bedrock depressions. However, due to similarities in $CH_4$ emissions and controls thereof, we also include all other lakes with organic-poor sediments within these classes. *Glacial Lakes* typically have rocky bottoms or mineral sediments with limited organic content. Lakes in this class are abundant on the Canadian Shield and in Fennoscandia, but can be found throughout the boreal and tundra biomes. Many *Glacial Lakes* have high

shoreline development index, with irregular, elongated shapes. Generally, *Glacial Lakes* are deeper than lakes in the other classes, when comparing lakes with similar lake area. *Glacial Lakes* have very low to moderate $CH_4$ emissions, with slightly greater emissions from *Small* than *Midsize Glacial Lakes,* driven by greater ebullitive emissions (Kuhn et al., 2021).





*Large Lakes* are greater than 10 km² in surface area. Most *Large Lakes* are glacial or structural/tectonic in origin. Lake
genesis is not considered for further differentiation within this land cover class. Emissions of $CH_4$ from *Large Lakes* are very
low to low (Kuhn et al., 2021).

### 2.2.3 River Classes

We include three river classes in BAWLD, *Large Rivers*, *Small Organic-Rich Rivers*, and *Small Organic-Poor Rivers*. *Large
Rivers* are described as 6th Strahler order rivers or greater, and generally have river widths >~75 m (Downing et al., 2012;
Lehner and Döll, 2004). *Small Organic-Rich Rivers* include all 1st to 5th order streams and rivers that drain peatlands or other
wetland soils, thus being associated with high concentrations of dissolved organic carbon and high supersaturation of $CH_4$.
Conversely, *Small Organic-Poor Rivers* drain regions with less wetlands and organic-rich soils, and generally have lower
concentrations of dissolved organic carbon and dissolved $CH_4$.

### 2.2.4 Other Classes

Four additional classes are included in BAWLD; *Glaciers, Rocklands, Dry Tundra,* and *Boreal Forests. Glaciers* include
both glaciers and other permanent snow and ice on land. *Rocklands* include areas with very poor soil formation and where
vegetation is largely absent. Rocky outcrops in shield landscapes, slopes of mountains, and high Arctic barren landscapes are
included in the class. The *Rocklands* class also includes artificial surfaces such as roads and towns. *Glaciers* and *Rocklands*
are largely considered to be neutral with respect to $CH_4$ emissions. The *Dry Tundra* class includes both lowland arctic tundra
and alpine tundra; both treeless ecosystems dominated by graminoid or shrub vegetation. *Dry Tundra* ecosystems generally
have near-surface permafrost, with seasonally thawed active layers between 20 and 150 cm depending on climate, soil
texture, and landscape position (van der Molen et al., 2007; Heikkinen et al., 2004). Near-surface permafrost in *Dry Tundra*
prevents vertical drainage, but lateral drainage ensures predominately oxic soil conditions. A water table is either absent or
close to the base of the seasonally thawing active layer. *Dry Tundra* is differentiated from *Permafrost Bogs* by having
thinner organic soil (<40 cm), and from *Tundra Wetlands* by their drained soils (average water table position >5 cm below
soil surface). *Dry Tundra* generally have net $CH_4$ uptake, but very low $CH_4$ emissions are also common (Kuhn et al., 2021).
*Boreal Forests* are treed ecosystems with non-wetland soils. Coniferous trees are dominant, but the class also includes
deciduous trees in warmer climates and landscape positions. *Boreal Forests* may have permafrost or non-permafrost ground,
where absence of permafrost often allow for better drainage. Overall, it is rare for anoxic conditions to occur in *Boreal
Forest* soils, and $CH_4$ uptake is prevalent, although low $CH_4$ emissions have been observed during brief periods during
snowmelt or following summer storms (Matson et al., 2009), or conveyed through tree stems and shoots (Machacova et al.,
2016). The *Boreal Forest* class also includes the few agricultural/pasture ecosystems within the boreal biome.





## 2.3 Expert assessment

Expert assessments can be used to inform various environmental assessments, and are particularly useful to assess levels of
uncertainty and to provide data that cannot be obtained through other means (Olefeldt et al., 2016; Loisel et al., 2021; Abbott et al., 2016; Sayedi et al., 2020). We solicited an expert assessment to aid in the modelling of fractional coverage of the 19 land cover classes within each BAWLD grid cell. Researchers associated with the Permafrost Carbon Network (www.permafrostcarbon.org) with expertise from wetland, lake, and/or river ecosystems within the BAWLD domain were invited to participate. We also included a few additional referrals to suitable experts outside the Permafrost Carbon Network.
A total of 29 researchers completed the expert assessment, and are included as co-authors of the BAWLD dataset. Each expert was asked to identify a region within the BAWLD domain for which they considered themselves familiar. Experts were then assigned 10 random cells from their region of familiarity and 10 cells distributed across the BAWLD domain that allowed for an overall balanced distribution of training cells (Figure S1). No cell was assessed more than once, and in total ~3% of the area of the BAWLD domain was included in the expert assessment. Each expert was asked to assess the percent
coverage of each of the 19 land cover classes within their 20 training cells. To guide their assessment, each expert was provided step-by-step instructions, plus information on the definitions of each land cover class, and a KML file with the data extracted from available spatial datasets for each grid cell (Table 1). Experts were asked to use their knowledge of typical wetland and lake classes within specific high-latitude regions, their ability to interpret satellite imagery as provided by Google Earth, and their judgement of the quality and relevance of available spatial datasets to make their assessments of
fractional cover. The information provided to experts to carry out the assessment is provided in the Supplementary Information.

## 2.4 Random forest model and uncertainty analysis

Random forest regression models were created to predict the percent coverage of all 19 individual BAWLD land cover classes, along with three additional models for total wetland, lake, and river coverage. Each land cover class was at first
modelled separately, which was followed by minor adjustments, described below, that ensured that the total land cover within each cell added up to 100%. All statistical analysis and modelling were done using R 4.0.2 (R Core Team, 2020), and the packages Boruta (v7.0.0; Miron et al., 2010), caret (v6.0-86; Kuhn, 2020), randomForest (v4.6-14; Liaw and Wiener, 2002), and factoextra (v1.0.7; Kassambara and Mundt, 2020).





**Table 2. Summary of random forest models for each land cover class in BAWLD.**

| Land cover classes | RMSE (%) | %Var | $m_{try}$[a] | Var.[b] | Relative Variable Importance[c] |
|---|---|---|---|---|---|
| Glaciers | 2.32 | 95.9 | 13 | 24 | GL30-ICE (100) NCS-GLA (6) |
| Rocklands | 9.79 | 67.2 | 21 | 41 | NCS-ROC (100) PZI-MTN (43) CAVM-BAR (39) PZI-RUG (24) CAPG-XLR (16) WC2-MAAP (14) WC2-MAAT (14) WC2-CMI (11) TEW-TUN (10) GLC2-FOR (10) |
| Tundra | 14.7 | 75.2 | 22 | 43 | TEW-TUN (100) TEW-BOR (41) PZI-PERM (26) GL30-TUN (16) WC2-MAAP (8) CAPG-CON (7) LAT (7) PZI-PERM (7) NCS-ROC(5) |
| Boreal Forest | 15.5 | 79.8 | 20 | 39 | GLC2-FOR (100) TEW-BOR (61) GL30-WET (23) TEW-TUN (21) GIEMS_MAMA (12) GIM_MAMI (11) GL30_TUN (10) |
| **Wetland Classes** | 8.5 | 85.8 | 25 | 48 | GL30-WET (100) NCS-HSE (46) NCS-HSO (44) PZI_FLAT (28) GWET-IN (10) WC2-MAAT (6) GLC2-WET (5) |
| Bog | 4.7 | 75.0 | 22 | 42 | NCS-HSO (100) GL30-WET (47) WC2-MAAT (23) PZI-PERM (17) PZI_FLAT (12) GLC2-WET (12) WC2-MAAP (6) |
| Fen | 4.5 | 76.3 | 21 | 40 | NCS-HSO (100) GL30-WET (56) WC2-MAAT (16) PZI-PERM (7) |
| Marsh | 1.3 | 54.1 | 18 | 34 | GL30-WET (100) GSW-OCC (80) GLC2-WET (56) BAS-RIV (31) NCS-HSO (17) WC2-MAAT (14) GLWD-RIV (13) PZI-PERM (12) IRYP-YED (10) GSW-RAR (10) |
| Permafrost Bog | 4.1 | 84.0 | 22 | 42 | NCS-HSE (100) CAPG-DIS (6) CAPG-XMF (5) |
| Tundra Wetland | 4.1 | 47.2 | 2 | 36 | CAVM-WET (100) GSW-OCC (95) NCS-HSE (80) CAPG-XHF (77) PZI-FLAT (63) GL30-TUN (61) WC2-CMI (57) HL-MID (57) IRYP-YED (56) LAT (53) |
| **Lentic Classes** | 2.03 | 97.8 | 32 | 32 | GL30-H2O (100) |
| Large Lake | 0.75 | 99.5 | 30 | 30 | HL-LAR (100) GL30-H2O (18) NCS-H2O (10) |
| Midsize Glacial Lake | 1.49 | 75.3 | 18 | 35 | HL-SHO (100) HL-MID (56) GSW-REG (18) GL30-H2O (8) NCS-H2O (8) GSW-PER (6) NCS-ROC (5) GLC2-WET (5) \ PZI-PERM (5) |
| Midsize Peatland Lake | 1.44 | 68.5 | 17 | 32 | HL-MID (100) NCS-HSO (24) GL30-WET (18) GWET-IN (18) HL-SHO (17) NCS-HSE (14) GSW-OCC (13) GLC2-WET (11) GIM-MAMI (9) GSW-PER (8) |
| Midsize Yedoma Lake | 0.86 | 68.4 | 16 | 31 | IRYP-YED (100) HL-MID (42) HL-SHO (23) CAVM-WET (14) CAPG-XHF (12) GSW-OCC (12) PZI-PERM (9) WC2-CMI (8) GL30-H2O (7) GSW-REG (6) |
| Small Glacial Lake | 0.89 | 15.6 | 2 | 29 | GSW-OCC (100) GSW-REG (81) HL-SHO (49) HL-MID (39) GIM-MAMA (37) GIM-MAMI (29) GSW-RAR (27) WC2-MAAT (26) PZI-PERM (25) GL30-H2O (23) |
| Small Yedoma Lake | 0.47 | 39.2 | 17 | 32 | IRYP-YED (100) WC2-MAAP (25) WC2-CMI (21) GLC2-H2O (15) GSW-OCC (14) CAVM-WET (14) PZI-FLAT (13) GSW_REG (11) CAPG-XHF (10) HL-MID (10) |
| Small Peatland Lake | 1.22 | 65.9 | 20 | 39 | GLC2-WET (100) GL30-WET (95) WC2-MAAT (34) CAPG-REL (32) NCS-HSE (26) PZI-PERM (24) NCS-HSO (23) GSW_OCC (23) LAT (17) WC2-CMI (16) |
| **Lotic Classes** | 0.49 | 90.3 | 16 | 31 | GLWD-RIV (100) BAS-RIV (39) GSW-OCC (11) |
| Large River | 0.48 | 90.4 | 17 | 32 | GLWD-RIV (100) BAS-RIV (39) GSW-OCC (10) |
| Small Org.-Poor Rivers | 0.09 | 18.7 | 2 | 41 | GSW-OCC (100) GLC2-WET (62) BAS-RIV(48) GSW-PER (42) GLWD-RIV (39) PZI-FLAT (38) GL30-H2O (33) GLC2-H2O (32) GIM-MAMA (32) WC2-MAAP (31) |
| Small Org.-Rich Rivers | 0.04 | 59.3 | 23 | 45 | GLC3-WET (100) PZI-FLAT (41) NCS-HSE (37) NCS-HSO (18) GLC2-WET (17) GWET-IN (13) GSW-OCC (12) CAPG-XLR (12) GLWD-RIV (11) BAS-RIV (11) |

[a]$m_{try}$ – is a fitted variable which decides how many variables that were randomly chosen at each split in the random forest analysis.
[b]Var. – indicates the number of variables that were included (out of 53) in the random forest analysis after the Boruta automatic feature selection.
[c]Relative variable importance – the most influential variable in the random forest analysis is assigned a 100% rating, and the importance of other variables are relative to this. See Table 1 for full descriptions of the variables. Here we list either all variable with >5% influence, or the top ten variables.



Prior to running the random forest analyses, we performed an automatic feature selection using a Boruta algorithm, (Miron et al., 2010). The Boruta algorithm completed 150 runs for each land cover class, after which subsets of the 53 possible data variables (Table 1) were deemed important and selected for inclusion in subsequent random forest models (Table 2). The

random forest models (Kuhn 2020, Liaw and Wiener, 2002) then used boot-strapped samples (i.e. the expert assessments of land cover fractional grid cell coverages) to grow 500 decision-trees ($n_{tree}$), with a subset of randomized data variables as predictors at each tree node ($m_{try}$). We used a 10-fold cross-validation with five repetitions providing $m_{try}$ as a tuneable parameter for model training. The random forest model output included the root mean squared error (*RMSE*), the percent of the expert assessment variability that was explained (*%Var*), and relative variable importance (Table 2). Relative variable

importance assigns a 100% importance to the variable with the most influence on the model, and then ranks all other variables relative to the influence of that variable. A bias correction (Song, 2015) was applied to the predicted data of land cover class coverages, as the models were found to overestimate low coverages and underestimate high coverages. After the bias correction, all bias-adjusted predictions <0% were set to 0%, while those >100% were set to 100% (for examples of the bias correction, see Figure S2). Next, we ensured that the combined coverage of all 19 land cover classes within each grid

cell added up to 100% by applying a proportional adjustment. In order to estimate the 5th and 95th percentile confidence bounds of the land cover predictions, we repeated the random forest analysis, as outlined above, an additional 20 times for each class. Each new run completely excluded 20% of the expert assessments, and the data were reshuffled four times. Each grid cell thus had 21 predictions of coverage for each of the 19 land cover classes, and for the cumulative wetland, lake, and river coverages, and the variability of these predictions were used to define the 5th and 95th percentile confidence bounds.


While each cell in BAWLD has a distinct land cover combination, we were also interested in identifying cells with similarities in their land cover compositions to distinguish between regions of the boreal and arctic domain that represent characteristic landscapes. We carried out a k-means clustering (Kassambara and Mundt, 2020) to group grid cells with similarities in their predicted land cover compositions. The k-means clustering was based on within-cluster sum of squares,

and we evaluated resulting maps with between 10 and 20 distinct classes. Using 15 clusters was deemed to balance within-cluster sum of squares and interpretability of the resulting map. As each cluster was defined largely by the relative dominance (or absence) of different wetland, lake, and river classes, we henceforth refer to these clusters as "wetscapes".

### 2.5 Evaluation against regional wetland datasets

We evaluated the predictions of wetland coverage in BAWLD against four independent, high-resolution regional land cover

datasets. These four datasets were chosen as they included more than one wetland class, thus enabling both evaluation against total wetland coverage and subsets of wetland classes. Two of these datasets were specifically aimed at mapping of wetlands, including Ducks Unlimited Canada's wetland inventories for western Canada as part of the Canadian Wetland Inventory (CWI; Canadian Wetland Inventory Technical Committee, 2016), and wetland mapping of the West Siberian Lowlands (WSL) (Terentieva et al., 2016). The other two datasets, the 2016 National Land Cover Database (NLCD) of



Alaska (Homer et al., 2020), and the 2018 CORINE Land Cover (Büttner, 2014) of northern Europe, represent more general land cover datasets. Data from these four datasets were summarized for each BAWLD grid cell where there was complete coverage. Data filtration was done for the CWI to remove cells if >10% of the cell was classified as burned, cloud, or shadow. There were few cases where there were equivalent wetland classes in BAWLD and these four regional datasets, and as such comparisons were generally made between groups of wetland classes that were considered generally comparable.

Similar evaluations were not possible for the lake classes, as there are no regional or circumpolar spatial datasets with information on lake genesis or sediment type.

## 3 Results and Discussion

The fractional land cover estimates of the Boreal-Arctic Wetland and Lake Dataset (BAWLD) is freely available online at https://doi.org/10.18739/A2C824F9X (Olefeldt et al., 2021), and includes both the central estimates and the 95% high and

low estimates of each land cover class in each grid cell.

### 3.1 Wetlands

Wetlands were predicted to cover a total of $3.2 \times 10^6$ km$^2$, or 12.5% of the BAWLD domain. The wetland area was dominated by *Fens* (29% of total wetland area), *Bogs* (28%), and *Permafrost Bogs* (27%), while *Marshes* and *Tundra Wetlands,* which have relatively higher CH$_4$ emissions, covered 5 and 12% of the wetland area, respectively (Table 3). This

estimate of total wetland area was greater than previously mapped within the BAWLD domain in GLC2 at $0.9 \times 10^6$ km$^2$ (Bartholomé and Belward, 2005), GL30 at $1.4 \times 10^6$ km$^2$ (Chen et al., 2015), and GWET at $2.3 \times 10^6$ km$^2$ (Matthews and Fung, 1987), but similar to the area of wetland soils in NCS (sum of "histosols", "histels", and "aqueous" soil coverage) at $3.0 \times 10^6$ km$^2$ (Hugelius et al., 2014). Differences between BAWLD and other estimates of wetland area likely stem partially from differences in wetland definitions, where e.g. definitions of wetlands in GLC2 and GL30 likely do not include wooded

bogs, fens, and permafrost bogs. While estimates of total wetland area GWET and in the NCS were closer to BAWLD, there were differences in the spatial distribution. Wetland cover in BAWLD was generally greater than in GWET and NCS in regions with low wetland cover. This likely reflects the ability of experts to infer the presence of small, or transitional wetlands that may otherwise be underestimated when mapped using other methodologies. Conversely, wetland cover in BAWLD was generally lower than in GWET and NCS in regions with high wetland cover. This was likely due to

differences in definitions, especially the exclusion of all open water ecosystems from wetlands in BAWLD. For example, it was common in the West Siberian Lowlands for the summed coverage of wetland soils in NCS and the "open water" coverage in GL30 to be substantially greater than 100%, suggesting that NCS included peatland pools and small ponds within its wetland soil coverage. Overall, the predictive random forest model of total wetland coverage was able to explain 86% of the variability in the expert assessments, and it was primarily influenced by the area of "wetlands" in GLC30 and the

wetland soil categories in NCS, followed by the coverage of "flat topography" in PZI (Gruber, 2012) (Table 2).





**Table 3. Summary of central estimates, 95% low and high confidence bounds, and the range of the 95% confidence interval expressed as a percent of the central estimate, for each of the land cover classes within the BAWLD domain.**

| Land cover classes | Central estimate ($10^6$ km$^2$) | Low confidence bound ($10^6$ km$^2$) | High confidence bound ($10^6$ km$^2$) | 95% confidence interval ($10^6$ km$^2$) | 95% CI (% of central estimate) |
|---|---|---|---|---|---|
| Glaciers | 2.09 | 1.99 | 2.21 | 0.22 | 11 |
| Rocklands | 2.74 | 2.21 | 3.40 | 1.19 | 44 |
| Tundra | 5.28 | 4.56 | 6.37 | 1.82 | 34 |
| Boreal Forest | 10.66 | 9.77 | 11.39 | 1.61 | 15 |
| **Wetlands** | 3.18 | 2.79 | 3.79 | 1.00 | 31 |
| Bog | 0.88 | 0.71 | 1.24 | 0.53 | 60 |
| Fen | 0.91 | 0.76 | 1.14 | 0.38 | 42 |
| Marsh | 0.16 | 0.12 | 0.23 | 0.11 | 71 |
| Permafrost Bog | 0.86 | 0.67 | 1.17 | 0.50 | 58 |
| Tundra Wetland | 0.38 | 0.31 | 0.53 | 0.22 | 59 |
| **Lakes** | 1.44 | 1.34 | 1.59 | 0.24 | 17 |
| Large Lake | 0.64 | 0.61 | 0.72 | 0.11 | 18 |
| Midsize Peatland Lake | 0.14 | 0.11 | 0.21 | 0.10 | 69 |
| Midsize Yedoma Lake | 0.034 | 0.023 | 0.071 | 0.05 | 140 |
| Midsize Glacial Lake | 0.38 | 0.33 | 0.43 | 0.10 | 26 |
| Small Peatland Lake | 0.12 | 0.085 | 0.17 | 0.08 | 71 |
| Small Yedoma Lake | 0.028 | 0.015 | 0.046 | 0.03 | 114 |
| Small Glacial Lake | 0.094 | 0.051 | 0.16 | 0.11 | 119 |
| **Rivers** | 0.12 | 0.094 | 0.19 | 0.10 | 81 |
| Large River | 0.080 | 0.072 | 0.11 | 0.04 | 50 |
| Small Organic-Rich Rivers | 0.010 | 0.005 | 0.054 | 0.05 | 502 |
| Small Organic-Poor Rivers | 0.033 | 0.020 | 0.067 | 0.05 | 143 |


The predictive random forest models for individual wetland classes differed both in terms of how much of the variability of the expert assessment data was explained, and in terms of which spatial data were most influential (Table 2). The model for *Permafrost Bog* coverage explained 84% of the variability in the expert assessments and was very strongly influenced by "histel" distribution in the NCS (Hugelius et al., 2014). Predictive models explained ~75% of the variability in the expert

assessments for *Bogs* and *Fens* separately (Table 2), but 87% when considered jointly. This shows that the available predictor variables were less suitable for modelling *Bogs* and *Fens* separately rather than jointly, which could partly be due to lower agreement among experts in assessments of *Bog* and *Fen* coverages compared to their sum. This would not be surprising as bogs and fens (and swamps) occur along hydrological and nutrient gradients, and can have vegetation characteristics that make them difficult to distinguish. Models for *Bogs* and *Fens* were both strongly influenced by the

"histosol" distribution in NCS, with secondary influences from the area of "wetlands" in GL30, "permafrost extent" in PZI, and "mean annual air temperature" in WC2. Predictive models for *Marsh* and *Tundra Wetlands* explained less of the





variability in expert assessments, at 54 and 47%, respectively. The predictive models for *Marsh* and *Tundra Wetlands* were influenced by variables that indicate a transition between terrestrial and aquatic ecosystems, e.g. area of "occasional inundation" in GSW, "rivers" in BAS, and "midsize lakes" in HL, but then differed in the influence of climate and

permafrost conditions.

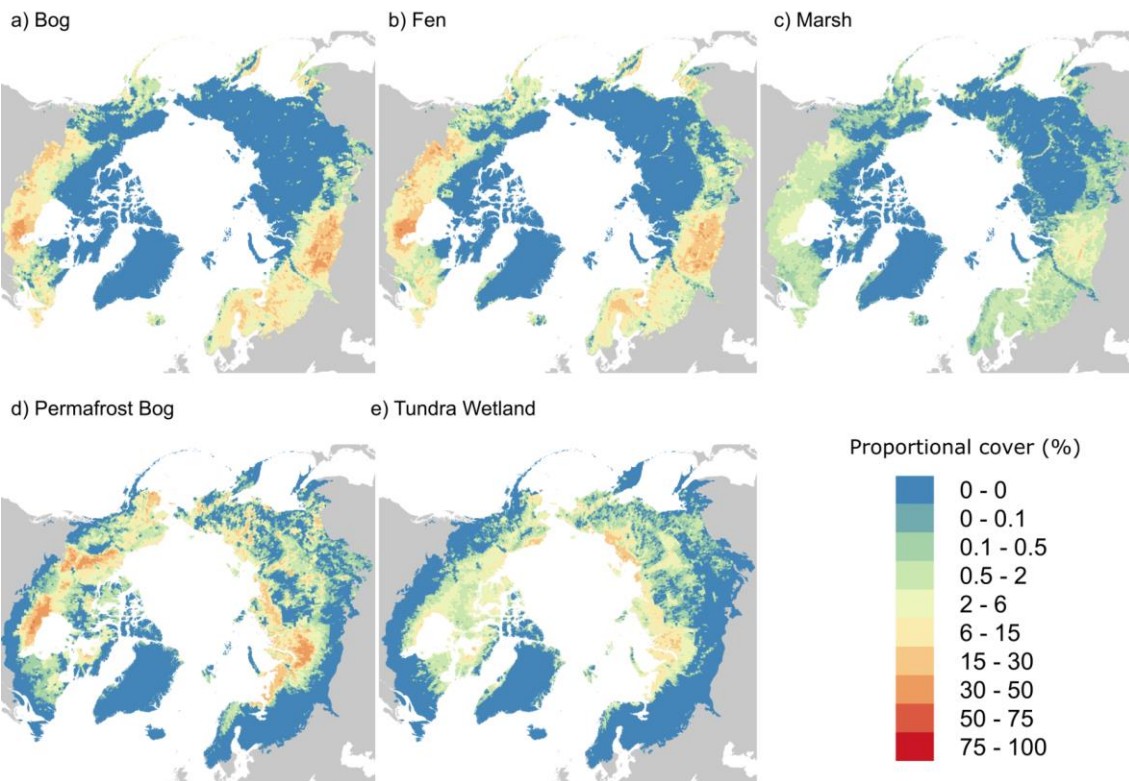

**Figure 2. Predicted distribution of wetland classes across the BAWLD domain; a) Bog, b) Fen, c) Marsh, d) Permafrost Bog, and e) Tundra Wetland.**


Each wetland class had a distinct spatial distribution (Figure 2). *Bogs* and *Fens* were the dominant wetland classes in relatively warmer climates, with high densities in the West Siberian Lowlands, Hudson Bay Lowlands, and the Mackenzie River Basin. While *Bogs* and *Fens* had similarities in their spatial distributions, there was also a relative shift in dominance from *Bogs* to *Fens* in relatively colder and drier climates (Figure S3). These trends are supported by bog to fen transitions

observed both within and between regions (Packalen et al., 2016; Vitt et al., 2000a; Väliranta et al., 2017), but may not be universal (Kremenetski et al., 2003). *Marshes* were also found in warmer climates and largely associated with *Bogs* and *Fens*, but with a more evenly spread distribution. The highest abundance of *Marsh* coverage was predicted for the Ob River floodplains, a region with very few studies of $CH_4$ emissions (Terentieva et al., 2019; Glagolev et al., 2011). *Bogs*, *Fens*, and *Marshes* all decreased in abundance in colder climates, with *Permafrost Bogs* becoming more abundant than *Bogs* when

mean annual temperatures were below -2.5°C, corresponding to findings from western Canada, Fennoscandia, and the West
Siberian Lowlands (Vitt et al., 2000b; Seppälä, 2011; Terentieva et al., 2016). *Tundra Wetlands* became dominant over *Fens*
and *Marshes* when mean annual air temperatures were below -5.5°C (Figure 3). *Tundra Wetlands* were predicted to be most
abundant in the lowland regions across the Arctic Ocean coast, with especially high abundance in northern Alaska, eastern
Siberia, and on the Yamal and Gydan Peninsulas in western Siberia.


**Figure 3. Relative abundance of the five wetland classes across a gradient of mean annual temperatures.**

We found good agreement between the distribution of wetlands in BAWLD and that of four independent regional spatial
datasets (Figure 4, Figure S4). The best agreements for total wetland cover were between BAWLD and the two datasets
dedicated specifically to wetland mapping; with $R^2$ of 0.76 with the WSL dataset and 0.72 with the CWI dataset. There were
also strong relationships between BAWLD and the WSL dataset for the distribution of specific wetland classes, for both
drier wetland classes ("ridge"+"ryam"+"palsa" vs. *Permafrost Bog + Bog*) and wetter classes ("fen" + "hollow" vs. *Fen*).
When comparing "wet hollow" of the WSL dataset and *Marsh* in BAWLD there were discrepancies, but they were primarily
attributed to the explicit exclusion of the Ob river floodplains in the WSL dataset (Figure S4). For the wettest classes, we had
only a weak relationship ($R^2$ = 0.19) between the CWI "Marsh" class and the sum of the BAWLD *Marsh* and *Tundra
Wetland* classes, but the overall average abundance for comparable grid cells was similar at 1.4 and 2.2%, respectively.
Agreements between BAWLD and the NLCD and CLC datasets were lower, especially for the relatively drier wetland
classes (Figure S4). Lower agreement between BAWLD and some classes of regional wetland datasets should not be
interpreted to demonstrate poor accuracy of BAWLD, as differences can be due to class definitions, large mapping units, and
relatively low accuracy of the non-wetland specific regional datasets.

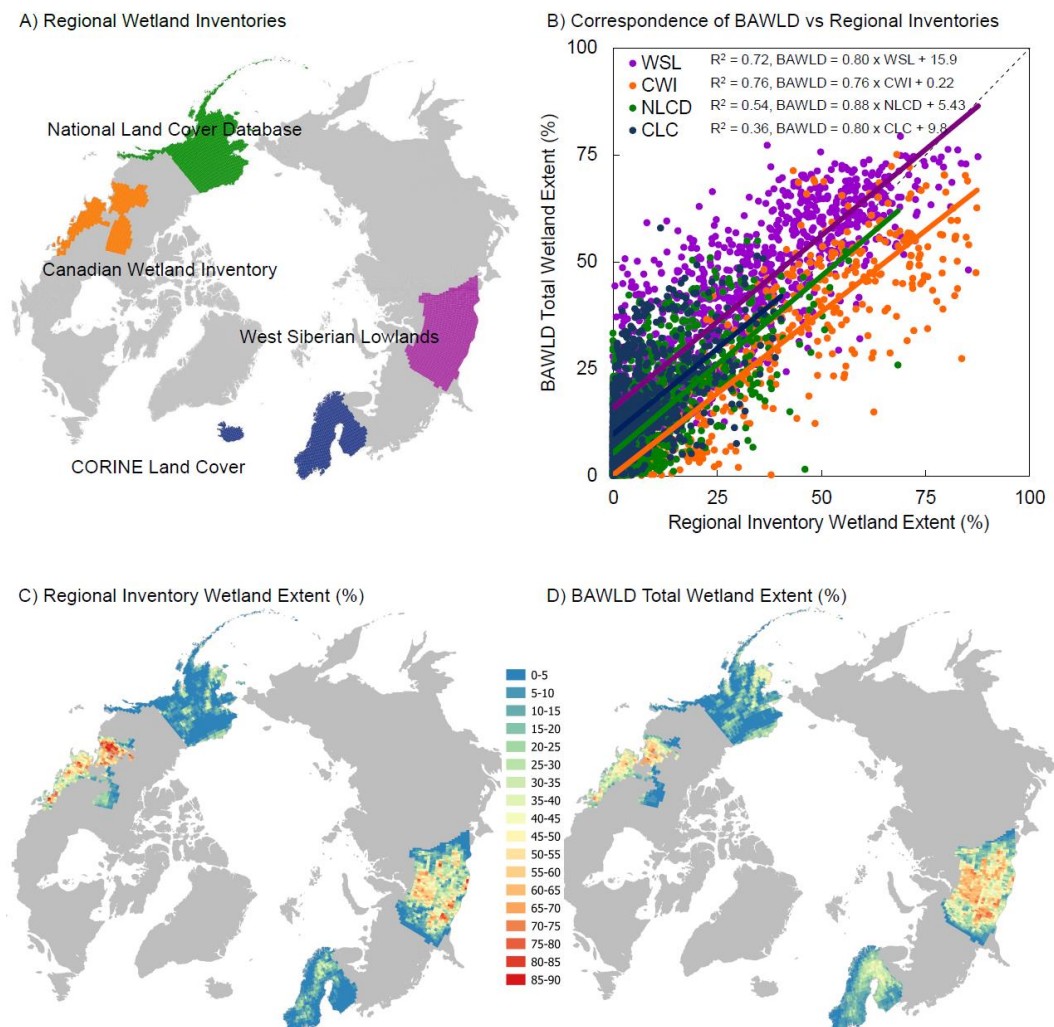

**Figure 4. Comparison of total wetland extent between BAWLD and four regional independent wetland inventories; the National Land Cover Database (NLCD), the Canadian Wetland Inventory (CWI), the wetland mapping of the West Siberian Lowlands (WSL), and the CORINE Land Cover dataset (CLC). a) Spatial extents of the regional datasets, b) Correlations between grid cell wetland coverages in BAWLD and the regional datasets, c) Spatial distribution of total wetland coverages in the four regional datasets, d) Spatial distribution of total wetland coverage in BAWLD for grid cells corresponding with the regional datasets.**

The 95% confidence intervals for predictions of abundance varied both between wetland classes and among regions (Table 3, Figure S5, S6). The confidence interval for total wetland area ranged between 2.8 and $3.8 \times 10^{6}$ km², i.e. a range that represented 31% of the central estimate. The range of the confidence interval depends both on how much consensus there is among experts in their assessments, and how well the available spatial datasets used in the random forest modelling can explain the expert assessments. The considerable range of the confidence interval for wetlands likely stems from a



combination of these two components. The confidence interval for total area of individual wetland classes varied between representing 42% (*Fens*) and 71% (*Marshes*) of respective central estimates. The absolute range of confidence intervals for individual cells generally increased with higher central estimates of abundances, but the range of confidence intervals decreased if expressed as a percent of the central estimate (Figure S7).

**3.2 Lakes**

Lakes were predicted to cover a total of 1.44 $\times 10^6$ km$^2$, or 5.6% of the BAWLD domain. *Large Lakes* had the greatest lake area (44% of total lake area), followed by *Midsize Glacial Lakes* (26%) and *Midsize Peatland Lakes* (10%) (Table 3). The lake classes with the highest CH$_4$ emissions, *Small Yedoma Lakes* and *Small Peatland Lakes*, jointly covered 10% of the total lake area. The total predicted lake area in BAWLD was higher than the area of lakes in HL (1.20 $\times 10^6$ km$^2$) which only includes lakes >0.1 km$^2$, and was similar to the area of "open water" in GL30 (1.43 $\times 10^6$ km$^2$). The "open water" class in the

GL30 dataset is, however, based on Landsat 30 m resolution data and thus excludes very small open water areas, while it includes both lentic and lotic open water. The 95% confidence interval for the total lake area in BAWLD was 0.24 $\times 10^6$ km$^2$, or 17% of the central estimate.

The predictive models for the three midsize lake classes each explained between 69 and 75% of the variability in expert

assessments, while a model for the sum of the three midsize lake classes explained 99.1%. The predictive model for the sum of the three midsize lake classes was almost exclusively influenced by the area of "midsize lakes" in HL, while the three midsize lake classes were differentiated through further influences by the area of "yedoma ground" (*Midsize Yedoma Lakes*), by the area of "histosols" and "histels" in NCS, and "wetlands" in GL30 (*Midsize Peatland Lakes*), and by "shoreline length" in HL (*Midsize Glacial Lakes*). The influence of "shoreline length" for *Midsize Glacial Lakes* shows that experts

associated glacial lakes with high shoreline development, and low shoreline development with peatland and yedoma lakes. Despite similarities in how much of the expert assessments were explained by the predictive models (69-75%), the extrapolation to the BAWLD domain led to large differences in the 95% confidence interval, which represented only 26% of the central estimate for *Midsize Glacial Lakes*, while representing 69 and 140% for *Midsize Peatland* and *Midsize Yedoma Lakes*, respectively (Table 3, Figure S8). *Midsize Glacial Lakes* were predominately predicted to have high abundances on

the Canadian Shield and in Fennoscandia, while *Midsize Yedoma Lakes* were associated with the lowland, coastal tundra regions of Northeast Siberia and Alaska, and *Midsize Peatland Lakes* were especially common in the West Siberian Lowlands, but also common in the peatland regions of the Hudson Bay Lowlands and the Mackenzie River Basin and in coastal lowland regions (Figure 5).



**Figure 5. Predicted distributions of lake and river classes within the BAWLD domain; a) Large Lakes, b) Midsize Glacial Lakes, c) Midsize Peatland Lakes, d) Midsize Yedoma Lakes, e) Small Glacial Lakes, f) Small Peatland Lakes, g) Small Yedoma Lakes. h) Large Rivers, i) Small Organic-Rich Rivers, j) Small Organic-Poor Rivers.**





*Small Glacial*, *Yedoma*, and *Peatland Lakes* were jointly estimated to cover 0.9% of the BAWLD domain. The predictive models explained 16, 39, and 66% of the variability, respectively (Table 2). The relatively lower predictive power for small

lakes was not unexpected, given lack of information on the smallest open water systems in the available spatial data, the variable abundance of very small open water systems among landscapes (Muster et al., 2019), and a lower relative consensus among experts when assessing classes with generally small fractional coverages. Models for all three small lake classes were influenced by the area of "occasional inundation" in GSW but were then differentiated by variables largely similar to those that were characteristic of the corresponding midsize lake classes (Table 2). The predicted distributions of the small lake

classes were also largely similar to that of the corresponding midsize lake type classes (Figure 5). The overall predicted area of small lakes was $0.24 \times 10^6$ km$^2$, representing 17% of the total lake area. The combined 95% uncertainty for the three classes ranged between 0.15 and $0.38 \times 10^6$ km$^2$ (Table 3, Figure S8), suggesting that small lakes represent between 11 and 26% of the total lake area. Previous assessments have estimated that open water ecosystems <0.1 km$^2$ represent between 21 and 31% of global lake area (Holgerson and Raymond, 2016), but relied on assumptions in the statistical modelling which

may lead to bias for boreal and arctic regions (Cael and Seekell, 2016; Muster et al., 2019).

### 3.3 Rivers

Rivers were predicted to cover a total of $0.12 \times 10^6$ km$^2$, or 0.47% of the BAWLD domain. *Large Rivers* accounted for 65% of the total river area in BAWLD. These estimates were similar to global assessments, where streams and rivers have been

estimated to cover between 0.30 and 0.56% of the land area, with 65% of the river area consisting of large rivers of 6$^{th}$ or greater stream order (Downing et al., 2012). The predictive model for *Large Rivers* was strongly influenced by the area of "large rivers" in GLWD, but experts consistently made lower assessments which led to an overall 15% lower area of *Large Rivers* compared to the area of rivers in GLWD within the BAWLD domain.

*Small Organic-Poor* and *Small Organic-Rich Rivers* were estimated to represent 27% and 8%, respectively, of the total river area. The predictive models for the *Small Organic-Poor* and *Organic-Rich Rivers* explained 19 and 59% of the expert assessments, and were distinctly influenced by the area of "occasional inundation" in GSW and "wetlands" in GLC30, respectively. The estimated area of small rivers varied among experts, reflecting difficulties in consistent assessments among experts for land cover classes with low extents (<1% in most grid cells). The distributions of expert assessments for small

river areas were non-normal, leading to a long upper tail for the 95% confidence interval (Figure S9). For example, the low, central, and high estimates for the area of *Small Organic Rich Rivers* were 0.005, 0.10 and $0.54 \times 10^6$ km$^2$, respectively. The predicted distributions showed that *Small Organic-Rich Rivers* was closely associated with the distribution of the BAWLD wetland classes, while *Small Organic-Poor Rivers* dominated elsewhere, with especially high abundances in regions with higher mean annual precipitation (Figure 5).





## 3.4 Other Classes

*Boreal Forest*, *Dry Tundra*, *Rocklands*, and *Glaciers* were predicted to cover 10.7, 5.3, 2.7 and 2.1 $\times 10^6$ km$^2$, respectively, within the BAWLD domain (Figure S10). The predictive models explained between 96% (*Glaciers*) and 67% (*Rocklands*) of the variability in expert assessments. While the predictive models for *Glaciers* was almost exclusively influenced by the area of "permanent snow and ice" in GL30, several variables influenced predictions of *Rocklands* – including area of "rocklands" in NCS, "mountainous" and "rugged" terrain in PZI, and "barrens" in CAVM. The predictive models for *Boreal Forest* and *Tundra* suggested that the transition between these classes was strongly influenced by the area "forest" in GLC2, and by the distinction between "tundra" and "boreal" terrestrial ecoregions in TEW.

## 3.5 Wetscapes

We defined "wetscapes" as regions with characteristic composition of specific wetland, lake, and river classes. Our clustering analysis distinguished 15 typical wetscapes within the BAWLD domain (Figure 6), each defined by the relative presence or absence of the 19 BAWLD classes (Table S1). Visualising the distribution of wetscapes provides information on regions that are likely to have similarities in the magnitude, seasonality, and climatic controls over CH$_4$ emissions.

Three wetscapes common in boreal regions were differentiated based on the abundance of non-permafrost wetlands. The *Sparse*, *Common*, and *Dominant Boreal Wetlands* wetscapes all had limited lake coverage (<6% on average), but had 15, 35, and 60% combined coverages of *Bogs*, *Fens*, and *Marshes*, respectively. The *Dominant Boreal Wetlands* wetscape was almost exclusive to the non-permafrost regions of the Hudson Bay Lowlands and the West Siberian Lowlands. The *Common Boreal Wetlands* wetscape was more widespread, found adjacent to the core areas of the Hudson Bay Lowlands and the West Siberian Lowlands, but also in the Mackenzie River Basin, northern Finland, European Russia, and in the Kamchatka Lowlands. The *Sparse Boreal Wetlands* wetscape was widespread in Sweden, Finland, European Russia, and the southern boreal regions of Canada outside of Yukon. Emissions of CH$_4$ from these regions are likely dominated by wetlands rather than lakes, with main sensitivity to climate change being altered water balance (Tarnocai, 2006; Olefeldt et al., 2017; Olson et al., 2013).



Figure 6. Wetscapes of the Boreal-Arctic Wetland and Lake Dataset. Wetscapes are defined by their characteristic composition of the BAWLD land cover classes, and thus groups regions with similar abundances (or absences) of specific wetland, lake, and river classes. The 15 wetscapes have their average land cover composition indicated by pie charts, with the legend shown in the bottom left. For clarity, the small and mid-sized lakes classes were combined for glacial, peatland, and yedoma lakes, and the river classes were omitted from the pie charts. No land cover pie charts are shown for the Large Lakes, Rivers, and Glaciers wetscapes.



The *Lake-rich Peatlands* and the *Permafrost Peatlands* wetscapes were both found in lowland regions with discontinuous permafrost, near the boreal to tundra transition. The *Lake-rich Peatlands* wetscape was almost exclusively found in the West Siberian Lowlands, north of the Ob River. This wetscape was characterized by roughly equal abundances of *Bogs*, *Fens* and

*Permafrost Bogs* (each 14-16%), along with 8% *Marshes*, 9% *Small Peatland Lakes* and 5% *Midsize Peatland Lakes*. It is notable that this wetscape, with the highest coverages of high-CH$_4$ emitting marshes and peatland lakes, has no presence in North America. The *Permafrost Peatlands* wetscape was conversely primarily found in the Hudson Bay Lowlands and the Mackenzie River Basin, with additional coverage along the Arctic Ocean coast in European Russia, in interior Alaska, and in the Anadyr Lowlands of far eastern Russia. This wetscape had the greatest abundance of *Permafrost Bogs* (27%), with less

contribution from other wetland classes (16%), and relatively low abundance of lakes (7%). The *Lake-rich Peatlands* wetscape likely has the highest regional CH$_4$ emissions, while the *Permafrost Peatlands* wetscape likely has low to moderate emissions. However, CH$_4$ emissions from both these wetscapes are likely highly sensitive to climate change due to the rapid ongoing and future permafrost thaw that causes expansion of thermokarst lakes and non-permafrost wetlands at the expense of *Permafrost Bogs* (Bäckstrand et al., 2008; Turetsky et al., 2002).


Three wetscapes were found in lowland tundra regions, and varied in relative dominance of different wetland and lake classes. *Wetland-rich Tundra* had 23% wetlands but only 7% lakes, and was found on the Gydan and Taymyr peninsulas in North Siberia, with minor extents in far eastern Siberia and in Alaska. *Wetland and Lake-rich Tundra* had similar wetland cover (24%) but twice the coverage of lakes (15%), split equally between glacial and peatland lakes. It was found on the

Alaska North Slope along with minor extents on the Yamal Peninsula, the Mackenzie River Delta, and on sections of Baffin Island. Lastly, the *Wetland and Lake-rich Yedoma Tundra* was characterized by the highest abundance of yedoma lakes (8%), a total wetland and lake coverage of 46%, and was primarily found in the Kolyma Lowlands, with minor extents in the Yukon-Kuskokwim Delta and on the Alaska North Slope. These regions may have sensitive CH$_4$ emissions, particularly associated with thermokarst lake expansion where highly labile yedoma sediments fuel high CH$_4$ production (Walter

Anthony et al., 2016).

The remaining seven wetscapes are likely to have overall low CH$_4$ emissions, or even net uptake, as a result either from the dominance of low-CH$_4$ emitting classes or due to the relative absence of wetland and lake classes. The *Dry Tundra* wetscape was common in regions of undulating topography of northernmost Siberia, the Alaska North Slope, and the western

Canadian Arctic, and was characterized by relatively low abundances of wetlands (9%) and lakes (3%). The *Lake-rich Shield* wetscape was exclusive to the Canadian Shield, and although it had a high abundance of lakes (18%), these were almost completely dominated by low-CH$_4$ emitting large lakes and glacial lakes. The *Upland Boreal* wetscape dominates boreal regions of Siberia but is also found in the Yukon, Alaska, and Quebec, and was defined by having <5% wetlands and 0.5% lakes. The *Alpine and Tundra Barrens* wetscape had <2% wetlands and ~1.5% lakes, and dominates the Greenland coast, the

high-latitude polar deserts of the Canadian Arctic Archipelago, and the mountain ranges in Fennoscandia, Alaska, Yukon,



and eastern Siberia. Lastly, the *Glaciers*, *Large Lakes*, and *Large Rivers* wetscapes were defined by the dominance of the namesake BAWLD classes.

## 5 Data Availability

The fractional land cover estimates from the Boreal-Arctic Wetland and Lake Dataset (BAWLD) is freely available at the
Arctic Data Center (Olefeldt et al., 2021): https://doi.org/10.18739/A2C824F9X. The dataset is provided as an ESRI shapefile (.shp) and as a Keyhole Markup Language (.kml) file.

## 6 Conclusions

The Boreal-Arctic Wetland and Lake Dataset (BAWLD) was developed to provide improved estimates of areal extents of five wetland classes, seven lentic ecosystem classes, and three lotic ecosystem classes by leveraging expert knowledge along
with available spatial data. By differentiating between wetland, lake and river classes with distinct characteristics, BAWLD will be suitable to support large-scale modelling of high-latitude hydrological and biogeochemical impacts of climate change. In particular, BAWLD has been developed with the aim to facilitate improved modelling of current and future $CH_4$ emissions. For example, a dataset of empirical $CH_4$ data was co-developed with BAWLD (Kuhn et al., 2021), ensuring that the land cover classification was meaningful for the separation of classes based on distinct magnitudes and controls of $CH_4$
emissions. By being based on an expert assessment and existing spatial dataset rather than a remote sensing approach, BAWLD was able to provide predictions for abundance of high-$CH_4$ emitting wetland and lake classes that have limited extents but disproportionate influences on regional and overall $CH_4$ emission. Using BAWLD for upscaling of $CH_4$ emissions will reduce issues of representativeness of empirical data for upscaling, reduce the risk of overlap between wetland and lake classes, and allow for more rigorous uncertainty analysis.

**Author contributions**

This study was conceived by DO. The GIS work was done by MH. The information sent to experts to complete the expert assessment was compiled by DO, MH, and MAK. All co-authors completed the expert assessment. The random forest modelling was led by DO, with input from TB, AR, and MJL. Data analysis and visualizations was led by DO with input from all co-authors. The manuscript was written by DO with contributions from all co-authors.

**Competing Interests**

The authors declare no competing interests.



## Acknowledgements

Financial support to DO was provided the National Science and Engineering Research Council of Canada (NSERC) Discovery grant (RGPIN-2016-04688) and the Campus Alberta Innovates Program. CT was supported by ERC (#851181) and the Helmholtz Impulse and Networking Fund. AM was supported by the Gordon and Betty Moore Foundation (Grant GBMF5439, 839; Stanford University). DB was supported by ERC (#725546), Swedish Research Council VR (#2016-04829), and FORMAS (#2018-01794). FJWP was supported by the Norwegian Research Council under grant agreement 274711, and the Swedish Research Council under registration no. 2017-05268. GG was supported through the BMBF KoPf Synthesis project (03F0834B). JDW was supported by NASA Earth Science (NNH17ZDA001N). MJL was supported by NSF-EnvE (#1928048). MS was supported by the Natural Sciences and Engineering Research Council of Canada (NSERC) through the Canada Research Chairs program. RKV was supported by the National Aeronautics and Space Administration IDS program (NASA grant NNX17AK10G). SAF was supported by the Natural Sciences and Engineering Research Council of Canada. SET was supported by funding from the Campus Alberta Innovates Program. Ducks Unlimited Canada's Wetland Inventories were funded by various partnering organizations: Environment and Climate Change Canada, Canadian Space Agency, Government of Alberta, Government of Saskatchewan, U.S. Forest Service, U.S. Fish and Wildlife Service, PEW Charitable Trusts, Canadian Boreal Initiative, Alberta-Pacific Forest Industries Inc., Mistik Management Ltd., Louisiana-Pacific, Forest Products Association of Canada, Weyerhaeuser, Lakeland Industry and Community, Encana, Imperial Oil, Devon Energy Corporation, Shell Canada Energy, Suncor Foundation, Treaty 8 Tribal Corporation ("Akaitcho"), and Dehcho First Nations. The Permafrost Carbon Network provided coordination support, and is funded by the NSF PLR Arctic System Science Research Networking Activities (RNA) Permafrost Carbon Network: Synthesizing Flux Observations for Benchmarking Model Projections of Permafrost Carbon Exchange, Grant # 1931333 (2019-2023).



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
