# Peer review of "The Boreal-Arctic Wetland and Lake Dataset (BAWLD)"

_Earth System Science Data, 2021_

## Author Comment (AC1)

**We thank the three reviewers for their comments and suggestions, which have helped improve the manuscript. Below, we respond point by point on the revisions we have done to address the reviewers' comments. We have further made changes to 2.2.1 and 2.2.2. where we describe and define the wetland and lake classes of BAWLD, in response to a comment from the handling editor. This manuscript describes the BAWLD land cover dataset, while the associated dataset with methane emissions (BAWLD-CH4) is described by Kuhn et al., (in review – ESSD). We initially included the same text on the descriptions and definitions of the land cover classes in both manuscripts, but we have now reduced the text in this manuscript. This manuscript now has shorter descriptions of the wetland and lake classes, and the reader is referred to Kuhn et al. (2021), for the details of the individual classes.**

Reviewer 1

Synopsis

The manuscript describes the creation of a new land cover classification system designed specifically for quantifying methane emissions from Arctic and Boreal regions. This science has been plagued by the complexity of methane biogeochemistry and the landscape heterogeneity of the region. Thus, estimates of northern high-latitude emissions carry large uncertainty. The authors have somewhat reinvented the wheel by designing a new model grid by combining expert knowledge and machine learning-based techniques to estimate fractional cover of 19 classes within a half-degree grid. They use schemes to harmonize land cover types to reduce double counting issues with bottom up emission budgeting. The large collaborative effort demonstrates that this technique could be widely accepted and highly useful for tracking methane emissions from the rapidly changing north. The article is well written and very near its final form. I can only recommend very minor changes to help improve the clarity and impact of the manuscript.

General Comment

This manuscript was written as a companion to Kuhn et al. 2021 (BAWLD-CH4: Methane Fluxes from Boreal and Arctic Ecosystems), which contains more quantitative methane information for the BAWLD model. Kuhn et al. 2021 is heavily cited in this manuscript, however a better connection to this paper could be made in the introduction and/or the conclusion. This would increase the impact and utility of both papers. A paragraph could be warranted to more clearly and explicityly illustrate/bridge how the two papers are companions to one another. This could also potentially improve on the lack of quantitative methane information in this manuscript. I was somewhat expecting a new pan-arctic/boreal annual CH4 estimate based on the BAWLD modeling in either this manuscript or Kuhn et al. 2021, however no global estimate was produced. This seems to be the potential a-priori gridded data that was suggested as highly useful to the inverse modelling community.

**Our Response:**

**We have added/revised a few sentences in the last paragraph of the introduction, and in the final paragraph of the conclusions to further describe how BAWLD and BAWLD-CH4 are related, and what data is presented in which article.**

**The relevant part of the introduction now reads:**

"Here we present the Boreal-Arctic Wetland and Lake Dataset (BAWLD), an expert knowledge-based land cover dataset. A companion dataset with chamber, and small-scale observations of CH4 emissions (BAWLD-CH4) is presented in Kuhn et al., (2021), and it uses the same land cover classes as BAWLD. The land cover classes were developed to distinguish between classes with distinct CH4 emissions, and include five wetland, seven lake, and three river classes."

The relevant part of the conclusions now read:

"The Boreal-Arctic Wetland and Lake Dataset (BAWLD) was developed to provide improved estimates of areal extents of five wetland classes, seven lentic ecosystem classes, and three lotic ecosystem classes by leveraging expert knowledge along with available spatial data. By differentiating between wetland, lake and river classes with distinct characteristics, BAWLD will be suitable to support large-scale modelling of high-latitude hydrological and biogeochemical impacts of climate change. In particular, BAWLD has been developed with the aim to facilitate improved modelling of current and future $CH_4$ emissions. For example, a companion dataset of empirical $CH_4$ data (BAWLD-CH4) (Kuhn et al., 2021) was co-developed with BAWLD, ensuring that the land cover classification was meaningful for the separation of classes based on distinct magnitudes and controls of $CH_4$ emissions. Future assessments of Boreal-Arctic $CH_4$ emissions based on combined use of the BAWLD and BAWLD-CH4 datasets will thus provide several refinements compared to previous bottom-up estimates. By being based on expert assessment and existing spatial dataset rather than a remote sensing approach, BAWLD was able to provide predictions for abundance of high-$CH_4$ emitting wetland and lake classes that have limited extents but disproportionate influences on regional and overall $CH_4$ emission (i.e. account for landscape $CH_4$ hotspots). Using BAWLD for upscaling of $CH_4$ emissions will reduce issues of representativeness of empirical data for upscaling, reduce the risk of overlap between wetland and lake classes, and allow for more rigorous uncertainty analysis."

We are currently working on a manuscript where we merge the BAWLD and BAWLD-CH4 datasets to produce estimates of current and future emissions under various scenarios. This will be presented in future manuscripts.

Specific Comments

Lines 181-184: Are the 53 variables available in all grid-cells? Do some grid cells contain more/less variables? Probably best to specify/clarify in the text.

Our response: We have added the following two sentences to section 2.1:

"High latitude data was not available for the GL30 (>82°N) and HL (>80°N) datasets and was coded as missing data. Regions outside the spatial extents of the CAVM, CAPG, and IRYP datasets were coded as 0, as it suggested absence of tundra vegetation, permafrost, and yedoma soils. "

Section 2.2.1: Could be really helpful to quickly provide some well-known real geographic examples for some of the main wetland classes.

Our response: We have reduced the length of text in section 2.2.1, the descriptions of the wetland classes in response to comment from the editor. The full description of the wetland classes is now found in Kuhn et al., (2021).

**It is difficult to describe distinct geographical examples for the wetland classes, as wetland classes generally often co-occur. For example, the large wetland regions of Hudson Bay Lowlands are a fine-scale mosaic of Bogs, Fens, Marshes, and Small Peatland Lakes, with Permafrost Bogs and Tundra Wetlands becoming more dominant in its northern regions. There are regional differences in the composition of wetlands, and BAWLD captures some of these known trends, e.g. the predominance of Bogs over Fens in southern boreal Finland and western Russia, while Fens are relatively more common than Bogs in boreal western Canada. Some of these regional differences in wetland composition among different regions is discussed in section 3.1 of the manuscript, and is a key aspect of the definitions of the "wetscapes".**

Line 280: Add comma after "As such"

**Our response: Done.**

Line 356: Add an "a" in between "have" and "high"

**Our response: Done.**

Line 442: Suggest changing the comma to a period and starting a new sentence with "We henceforth…"

**Our response: We have changed the sentence to read: "We henceforth refer to these clusters as "wetscapes", as each cluster was defined largely by the relative dominance (or absence) of different wetland, lake, and river classes.**

Figure 6: Minor suggestion: add the word "Legend" to the color key to quickly differentiate this wheel from the others. Maybe in the center? Bold? Maybe with a box around it? Take it or leave it, but it is confusing at first. At first glance, I interpreted the legend as a global distribution.

**Our response: We have given a bit more space for the pie chart acting as a legend, and we have written out "Legend" above it. It is also stated in the figure text that the pic chart in the bottom left is the legend for the other pic charts.**

Reviewer 2

In this manuscript, the authors presented a new dataset (BAWLD) of wetlands, lakes, rivers, and other land-cover types for the boreal and arctic areas. Although many land cover data have been proposed, I agree that this dataset has advantages in its comprehensiveness and expert assessment. Namely, this dataset would surely contribute to improve accuracy of methane emissions from this region, especially in terms of separation of wetland and freshwater sources. Therefore, I found enough merits to publish this manuscript.

I have two minor caveats. First, I could not understand the reason why the authors chose the spatial resolution of 0.5 degree? I know this resolution has been standard for global terrestrial models, but it may be difficult to capture spatial heterogeneity due to topography and micro hydrometeorology in this area. Indeed, several land-cover maps such as GLC2000 (used as an input data of random forest model in this study) have a spatial resolution of about 1km. One possible option may be to provide several data files with different spatial resolutions: e.g., 1km as a full resolution and 0.5 degree as an aggregated resolution. Second, I know that the Global Lake and Wetland Dataset (GLWD, Lehner and Döll 2004),

which contains multiple types of wetlands and lakes, has been used in several studies. However, the authors rarely mentioned about this dataset and used it only for river detection. For example in Fig. 4 and 4S, the authors did not include the GLWD into their inter-data comparison. I recommend making a comparison or discussion with the GLWD (and other data, if necessary) to clarify the advantage of the BAWLD. For example, the explicit separation of characteristic types such as permafrost wetlands and yedoma lakes look a clear advantage for data user working in this area.

**Our response: We chose to work with 0.5° grid cells as it is a standard resolution for global models, and because it avoids depicting a level of spatial detail that I do not think our approach can support, given the very different spatial resolutions, polygon sizes, and accuracy of the datasets used in the random forest regression models. We can not at this time change the spatial resolution of the dataset, as the random forest regression models need the expert assessment inputs as response variables, and all assessments were made using 0.5° grid cells. Using 0.5° grid cells allowed us to cover ~3% of the Boreal-Arctic domain with an expert assessment. Reducing the size of the grid cells would not have increased the number of cells assessed by the experts, and thus the models would be based on an assessment of a much smaller portion of the domain.**

**We did not use the GLWD data of wetland extent as the data GLWD uses is also used as an input for the Northern Circumpolar Soils Carbon Database. As such the GLWD is not independent from BAWLD, and a comparison is somewhat circular. GLWD also does not include a consistent wetland classification across countries, but instead relies on wetland classes from several different national inventories, so we would not be able to make a similar comparison as was done against the Canadian Wetland Inventory or the West Siberian Lowlands inventory.**

The manuscript gave full descriptions of the dataset. Although descriptions of individual wetland and lake types look lengthy, it may be useful for data users. Similarly, the authors provided a plenty of figures and descriptions as the supplementary file. The wetscape, derived from the wetland and lake data, can be excessive and unnecessary, but I agree that it is implicative. Finally, I recommend the manuscript is acceptable after minor revisions.

**Our response: As per suggestion from the editor, we have reduced the length of the descriptions of the wetland and lake classes. These descriptions are now found only in Kuhn et al., 2021.**

Technical points

Page 19 Line 13: Please note that Bohn et al. (2015) conducted a model intercomparison study on CH4 emissions in the West Siberia Lowland including the Ob River floodplains.

**Our response: In this sentence we were referring to field studies of measured CH4 emissions on the Ob floodplains. We have added the word "field" to the sentence:**

**"The highest abundance of *Marsh* coverage was predicted for the Ob River floodplains, a region with very few field studies of $CH_4$ emissions (Terentieva et al., 2019; Glagolev et al., 2011)."**

Page 30 Line 748: Several records in References lack the information on journal name. For example, Bastviken et al. (2004) was published from Global Biogeochemical Cycles. Please check also other records.

**Our response: Thanks for catching this, seems like an issue with my citation manager. All journal titles are now included in the reference-list.**

Reviewer 3

The manuscript addresses the very important topic of estimating the extent of different wetland types in northern latitude regions. As the authors mention, these regions are particularly affected by climate change and accurately estimating the extent of wetland types can help reduce the uncertainty associated with methane emissions, which is currently very high. In summary, the authors produce a state-of-the-art dataset that can be readily used by experimental scientists and modelers from various disciplines. The paper is also well structured and overall easy to follow. I recommend publication after minor revisions, especially given the impact of the work.

The only major point that I suggest the authors is to discuss more thoroughly is the spatial resolution of the dataset. 0.5x0.5 is relatively coarse, while land surface models are now moving towards much finer resolutions. So, I think it would be beneficial to discuss more what were the limiting factors for having to work with this resolution. This might inform next steps that can be taken in order one day have a finer resolution gridded dataset.

**Our response (from above, where same question was raised): Our response: We chose to work with 0.5° grid cells as it is a standard resolution for global models, and because it avoids depicting a level of spatial detail that I do not think our approach can support, given the very different spatial resolutions, polygon sizes, and accuracy of the datasets used in the random forest regression models. We can not at this time change the spatial resolution of the dataset, as the random forest regression models need the expert assessment inputs as response variables, and all assessments were made using 0.5° grid cells. Using 0.5° grid cells allowed us to cover ~3% of the Boreal-Arctic domain with an expert assessment. Reducing the size of the grid cells would not have increased the number of cells assessed by the experts, and thus the models would be based on an assessment of a much smaller portion of the domain.**

I also suggest expanding Table 1. More information could be included in this Table, such as the spatial and temporal resolution of the data sources.

**Our response: We have added information on spatial resolution in Table 1. Several data sources were based on polygons of variable areas, which is indicated. The spatial extent of the data sources which did not cover the whole BAWLD domain is described at the end of section 2.1:**

**"High latitude data was not available for the GL30 (>82°N) and HL (>80°N) datasets and was coded as missing data. Regions outside the spatial extents of the CAVM, CAPG, and IRYP datasets were coded as 0, as it suggested absence of tundra vegetation, permafrost, and yedoma soils. "**

One aspect that could be made clearer is how the expert assessment was integrated with the overall procedure to derive the dataset. How was this information used? For example, it was not clear (at least to me) whether this information was integrated into the modeling or not. The authors could expand a bit section 2.3 and provide more details in the introduction.

**Our response: The regression models explained in section 2.4 use the expert assessment of land cover coverage as the response variables. Hence the resulting BAWLD map can be thought of as a spatial**

**extrapolation of the expert assessment. To make this clear, we have added a sentence early in section 2.4:**

**"The regression models used the expert assessment of land cover coverage as the response variables."**

Lastly, I wonder what the authors think of other machine learning approaches that can take advantage of relatively high-resolution satellite images (computer visions tasks using convolution neural networks) and whether these approaches might prove useful to improve spatial resolution and details of BAWLD or similar databases.

**Our response: I have no doubt that high resolution satellite imagery and machine learning will be used to create high resolution maps of wetlands for large regions in the coming years. However, their accuracy will critically depend on ground-truth observations throughout their spatial domains. Wetlands have regional characteristics, in terms of dominant vegetation and dominant wetland landforms, and this is something that requires local expert knowledge. Hence, even as capabilities for working with high-resolution imagery and AI approaches grow in the coming years, I think there is a key role for expert assessments and input to ensure quality of future land-cover datasets.**